# SETTING THE RECORD STRAIGHT ON TRANSFORMER OVERSMOOTHING

## ABSTRACT

Transformer-based models have recently become wildly successful across a diverse set of domains. At the same time, recent work has shown that Transformers are inherently low-pass filters that can oversmooth the input, reducing the expressivity of their representations. A natural question is: How can Transformers achieve success given this shortcoming? In this work we show that in fact Transformers are not inherently low-pass filters. Instead, whether Transformers oversmooth or not depends on the eigenspectrum of their update equations. Our analysis extends prior work in oversmoothing and in the closely-related phenomenon of rank collapse. We show that many successful Transformer models have attention and weights which satisfy conditions that avoid oversmoothing. Finally, we validate our observations with a simple way to parameterize the weights of the Transformer update equations that allows for control over its spectrum, ensuring that oversmoothing does not occur. Compared to other solutions for oversmoothing, our approach does not require a new architecture, or any additional hyperparameters.

## 1 INTRODUCTION

In recent years, Transformer models Vaswani et al. (2023) have achieved astounding success across vastly different domains: e.g., vision Dosovitskiy et al. (2021); Touvron et al. (2021a), NLP Touvron et al. (2023); Wei et al. (2023); Kaddour et al. (2023), chemistry Schwaller et al. (2019). However, without massive training datasets, their performance can quickly saturate as model depth increases Kaplan et al. (2020); Wang et al. (2022)

This appears to be caused by a fundamental property of Transformer models: a recent line of work argues that they are inherently low-pass filters Wang et al. (2022); Park & Kim (2022); Guo et al. (2023); Ali et al. (2023). This causes them to oversmooth as the number of layers increases, eventually causing the representation of all features to converge to the same vector.

In practice, this has led to a search for replacements for self-attention layers, including completely new attention blocks Wang et al. (2022); Ali et al. (2023) and normalization layers Guo et al. (2023), convolutional layers Park & Kim (2022), fully-connected layers Liu et al. (2021a); Kocsis et al. (2022); Yu et al. (2022a), and even average pooling layers Yu et al. (2022b).

In this work we show that in fact, *Transformer models are not inherently low-pass filters*. Specifically, by analyzing the spectrum of the Transformer update equations, it is possible to place conditions on the eigenvalues of attention and weight matrices such that oversmoothing does not occur. We make the following contributions:

- We give a new characterization of how the eigenspectrum of attention and weight matrices in the Transformer update affects oversmoothing as depth increases. This generalizes prior work which analyzed the spectrum of attention matrices alone Wang et al. (2022); Ali et al. (2023).

- We show that a majority of the attention and weights of successful pre-trained Transformer models Dosovitskiy et al. (2021); Touvron et al. (2021a) have eigenvalues that satisfy conditions that avoid oversmoothing.

- We detail how the closely-related phenomenon of 'rank-collapse' Dong et al. (2021); Noci et al. (2022) will occur except in extremely rare cases. This extends prior analyses on

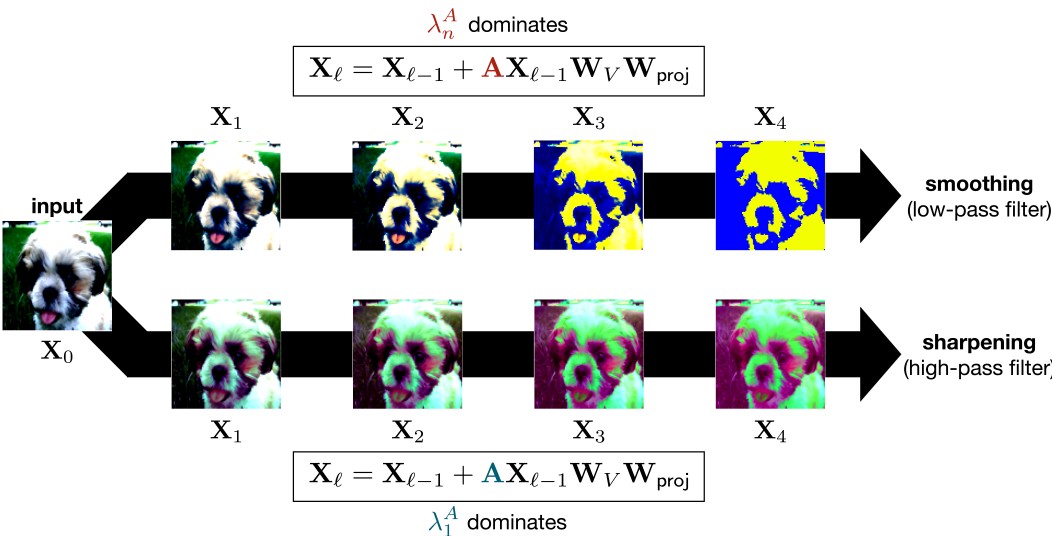

Figure 1: **Transformers can oversmooth, or not.** Evolution of the system $\mathbf{X}_\ell = \mathbf{X}_{\ell-1} + \mathbf{A}\mathbf{X}_{\ell-1}\mathbf{W}_V\mathbf{W}_{\mathsf{proj}}$. When $\mathbf{A}$ has its most positive eigenvalue dominate $\lambda_1^A$ we get oversmoothing. If instead its smallest eigenvalue dominates $\lambda_n^A$, we get sharpening.

rank collapse Dong et al. (2021), answering an open question about the role of the residual connection on rank-collapse.

- We describe a simple way to reparameterize the weights of the Transformer update to ensure that oversmoothing does not occur. Compared to other solutions to oversmoothing, our approach does not require a new architecture, or any additional hyperparameters.

## 2 BACKGROUND

**The Transformer Update.** At their core, Transformers are a linear combination of a set of 'heads'. Each head applies its own self-attention function on the input $\mathbf{X} \in \mathbb{R}^{n \times d}$ as follows

$$\mathbf{A} := \mathsf{Softmax}\Big(\frac{1}{\sqrt{k}}\mathbf{X}\mathbf{W}_Q\mathbf{W}_K^\top\mathbf{X}^\top\Big), \tag{1}$$

where the $\mathsf{Softmax}(\cdot)$ function is applied to each row individually. Further, $\mathbf{W}_Q, \mathbf{W}_K \in \mathbb{R}^{d \times k}$ are learned query and key weight matrices. This 'attention map' $\mathbf{A}$ then transforms the input to produce the output of a single head

$$\mathbf{A}\mathbf{X}\mathbf{W}_V\mathbf{W}_{\mathsf{proj}},$$

where $\mathbf{W}_V, \mathbf{W}_{\mathsf{proj}} \in \mathbb{R}^{d \times d}$ are learned value and projection weights. A residual connection is added to produce the output $\mathbf{X}_\ell$ of any layer $\ell$:

$$\mathbf{X}_\ell := \mathbf{X}_{\ell-1} + \mathbf{A}_\ell\mathbf{X}_{\ell-1}\mathbf{W}_{V,\ell}\mathbf{W}_{\mathsf{proj},\ell}, \tag{2}$$

It is possible to introduce further complexity by learning additional heads (i.e., additional $\mathbf{A}, \mathbf{W}_V$) and summing all head outputs. For simplicity we will describe properties of the single-head Transformer.

**Oversmoothing via Low-Pass Filtering.** There are many ways to measure oversmoothing, we opt here for the definition described in Wang et al. (2022) based on filtering, which we found the most intuitive. The overall idea is that we can view the layers of a deep learning model as a filtering operation that is applied repeatedly to $\mathbf{X}$. If the filtering operation is *low-pass*, it amplifies only the lowest frequency of $\mathbf{X}$, smoothing $\mathbf{X}$. On the other hand, a *high-pass* filter will amplify all other frequencies.

Specifically, let $\mathcal{F} : \mathbb{R}^{n \times d} \to \mathbb{C}^{n \times d}$ be the Discrete Fourier Transform (DFT). The DFT of $\mathbf{X}$ can be computed via matrix multiplication: $\mathcal{F}(\mathbf{X}) := \mathbf{F}\mathbf{X}$, where $\mathbf{F} \in \mathbb{C}^{n \times n}$ is equal to

$\mathbf{F}_{k,l} := e^{2\pi i(k-1)(l-1)}$ for all $k, l \in \{2, \ldots, n\}$ (where $i := \sqrt{-1}$), and is $1$ otherwise (i.e., in the first row and column). Define the Low Frequency Component (LFC), also called the Direct Current, of $\mathbf{X}$ as $\text{LFC}[\mathbf{X}] := \mathbf{F}^{-1}\text{diag}([1, 0, \ldots, 0])\mathbf{F}\mathbf{X} = (1/n)\mathbf{1}\mathbf{1}^{\top}\mathbf{X}$. Further, define the High Frequency Component (HFC), also called the Alternating Current, of $\mathbf{X}$ as $\text{HFC}[\mathbf{X}] := \mathbf{F}^{-1}\text{diag}([0, 1, \ldots, 1])\mathbf{F}\mathbf{X} = (\mathbf{I} - (1/n)\mathbf{1}\mathbf{1}^{\top})\mathbf{X}$.

**Definition 1** (Wang et al. (2022)). *Given an endomorphism $f : \mathbb{R}^n \to \mathbb{R}^n$ where $f^L$ denotes applying $f$ repeatedly $L$ times, $f$ is a low-pass filter if and only if for all $\mathbf{X} \in \mathbb{R}^{n \times d}$*

$$\lim_{L \to \infty} \frac{\|\text{HFC}[f^L(\mathbf{X})]\|_2}{\|\text{LFC}[f^L(\mathbf{X})]\|_2} = 0.$$

# 3 DO TRANSFORMERS ALWAYS OVERSMOOTH?

This is the main question we are trying to answer: are Transformers inherently low-pass filters, oversmoothing as depth increases? We will first introduce preliminary notation and assumptions. We then analyze how the eigenvalues of the Transformer update equations converge as depth increases. This allows us to derive convergence results for the features as well. Finally, using Definition 1, we derive the conditions under which Transformers are and are not low-pass filters.

## PRELIMINARIES

We start by rewriting eq. (2) to simplify the analysis. First note that Definition 1 applies for a fixed function $f$, applied $L$ times. Because of this, we will analyze applying same attention and weights at each layer[1]. Define the $\text{vec}(\mathbf{M})$ operator as converting any matrix $\mathbf{M}$ to a vector $\mathbf{m}$ by stacking its columns. We can rewrite eq. (2) vectorized as follows

$$\text{vec}(\mathbf{X}_\ell) = (\mathbf{I} + \underbrace{\mathbf{W}_{\text{proj}}^{\top}\mathbf{W}_V^{\top}}_{:=\mathbf{H}} \otimes \mathbf{A})\text{vec}(\mathbf{X}_{\ell-1}). \tag{3}$$

This formulation is especially useful because $\text{vec}(\mathbf{X}_L) = (\mathbf{I} + \mathbf{H} \otimes \mathbf{A})^L \text{vec}(\mathbf{X})$. We now introduce an assumption on $\mathbf{A}$ that is also used in Wang et al. (2022).

**Assumption 1** (Wang et al. (2022)). *The attention matrix is positive, i.e., $\mathbf{A} > 0$, and invertible.*

This assumption nearly always holds unless $\mathbf{A}$ numerically underflows. Initialization for $\mathbf{W}_Q, \mathbf{W}_K$ and normalization for $\mathbf{X}$ are often designed to avoid this scenario. In our experiments we never encountered $a_{ij} = 0$ for any element $(i, j) \in \mathbb{R}^n \times \mathbb{R}^n$ or $\mathbf{A}$ that was not invertible, in any architecture.

Note $\mathbf{A}$ is also right-stochastic, i.e., $\sum_j a_{i,j} = 1$, by definition in eq. (1). This combined with Assumption 1 immediately implies the following proposition.

**Proposition 1** (Meyer & Stewart (2023)). *Given Assumption 1, all eigenvalues of $\mathbf{A}$ lie within $(-1, 1]$. There is one largest eigenvalue that is equal to $1$, with corresponding unique eigenvector $\mathbf{1}$. No eigenvectors of $\mathbf{A}$ are equal to $0$.*

We leave the proof to the Appendix. We can now analyze the eigenvalues of the Transformer update equations.

## THE EIGENVALUES

As the number of layers $L$ in the Transformer update eq. (3) increases, one eigenvalue of $(\mathbf{I} + \mathbf{H} \otimes \mathbf{A})^L$ will dominate the rest (except in cases of ties, which is rare as $\mathbf{H}$ and $\mathbf{A}$ are learned). We define this formally below.

**Definition 2.** *At least one of the eigenvalues of $(\mathbf{I} + \mathbf{H} \otimes \mathbf{A})$, i.e., $(1 + \lambda_j^H \lambda_i^A)$ has a larger magnitude than all others, i.e., there exists $j^*, i^*$ (which may be a set of indices if there are ties) such that $|1 + \lambda_{j^*}^H \lambda_{i^*}^A| > |1 + \lambda_{j'}^H \lambda_{i'}^A|$ for all $j' \in \{1, \ldots, d\} \setminus j^*$ and $i' \in \{1, \ldots, n\} \setminus i^*$. These eigenvalues are called **dominating**.*

---

[1]It is straightforward to extend the analysis to changing attention and weight matrices with additional notation and conditions. We opted against doing this to make the results simpler and clearer. We plan to analyze this in future work.

Which eigenvalue(s) dominate will control the smoothing behavior of the Transformer.

**Theorem 1.** *Consider the Transformer update with fixed $\mathbf{A} > 0, \mathbf{H} := \mathbf{W}_{\mathsf{proj}}^\top \mathbf{W}_V^\top$, as described in eq. (3). Let $\{\lambda_i^A, \mathbf{v}_i^A\}_{i=1}^n$ and $\{\lambda_j^H, \mathbf{v}_j^H\}_{j=1}^d$ be the eigenvalue and eigenvectors of $\mathbf{A}$ and $\mathbf{H}$. Let the eigenvalues (and associated eigenvectors) be sorted as follows, $\lambda_1^A \leq \cdots \leq \lambda_n^A$ and $|1 + \lambda_1^H| \leq \cdots \leq |1 + \lambda_d^H|$. Let $\varphi_1^H, \ldots, \varphi_d^H$ be the phases of $\lambda_1^H, \ldots, \lambda_d^H$. As the number of layers $L \to \infty$, one eigenvalue dominates the rest (multiple dominate if there are ties):*

$$
\begin{cases}
\left.\begin{array}{ll}
(1 + \lambda_d^H \lambda_n^A) & \text{if } |1 + \lambda_d^H \lambda_n^A| > |1 + \lambda_d^H \lambda_1^A| \\
(1 + \lambda_1^H \lambda_1^A) & \text{if } |1 + \lambda_d^H \lambda_n^A| < |1 + \lambda_d^H \lambda_1^A|
\end{array}\right\} & \text{if } \lambda_1^A > 0 \\
\left.\begin{array}{ll}
(1 + \lambda_d^H \lambda_n^A) & \text{if } |1 + \lambda_d^H \lambda_n^A| > |1 + \lambda_k^H \lambda_1^A| \\
(1 + \lambda_k^H \lambda_1^A) & \text{if } |1 + \lambda_d^H \lambda_n^A| < |1 + \lambda_k^H \lambda_1^A|
\end{array}\right\} & \text{if } \lambda_1^A < 0, \varphi_d^H \in [-\tfrac{\pi}{2}, \tfrac{\pi}{2}] \\
\left.\begin{array}{ll}
(1 + \lambda_d^H \lambda_n^A) & \text{if } |1 + \lambda_d^H \lambda_n^A| > |1 + \lambda_d^H \lambda_1^A| \\
(1 + \lambda_d^H \lambda_1^A) & \text{if } |1 + \lambda_d^H \lambda_n^A| < |1 + \lambda_d^H \lambda_1^A|
\end{array}\right\} & \text{if } \lambda_1^A < 0, \varphi_d^H \in (\tfrac{\pi}{2}, \pi] \cup [-\pi, -\tfrac{\pi}{2})
\end{cases}
$$

*where $\lambda_k^H$ is the eigenvalue with the largest index $k$ such that $\varphi_k^H \in (\pi/2, \pi] \cup [-\pi, -\pi/2)$.*

The proof of Theorem 1 is in the Appendix. This Theorem says that depending on the phases (also called the 'arguments' or 'angles') of the eigenvalues of $\mathbf{H}$ and whether the eigenvalues of $\mathbf{A}$ are all positive (i.e., $\lambda_1^A > 0$) or some are negative (i.e., $\lambda_1^A < 0$, recall that by Proposition 1, $\mathbf{A}$ has no 0 eigenvalues), the dominating eigenvalue changes, and always contains either $\lambda_n^A$ or $\lambda_1^A$.

If the eigenvalues of $\mathbf{H}$ are all real, we can simplify Theorem 1, we detail this in Appendix B. Given a characterization of the eigenvalues as depth increases, we can identify how the features $\mathbf{X}_L$ converge.

THE FEATURES

**Theorem 2.** *As the number of total layers $L \to \infty$, the feature representation $\mathbf{X}_L$ converges. Which representation it converges to depends on the dominating eigenvalue, as given in Theorem 1. If a single eigenvalue dominates, there are two cases: (1) If $(1 + \lambda_j^H \lambda_n^A)$ dominates then,*

$$
\mathbf{X}_L \to (1 + \lambda_j^H \lambda_n^A)^L s_{j,n} \mathbf{1} \mathbf{v}_j^{H\top}, \tag{4}
$$

*(2) If $(1 + \lambda_j^H \lambda_1^A)$ dominates then,*

$$
\mathbf{X}_L \to (1 + \lambda_j^H \lambda_1^A)^L s_{j,1} \mathbf{v}_1^A \mathbf{v}_j^{H\top} \tag{5}
$$

*where $s_{j,i} := \langle \mathbf{v}_{j,i}^{Q^{-1}}, \mathsf{vec}(\mathbf{X}) \rangle$ and $\mathbf{v}_{j,i}^{Q^{-1}}$ is row $ji$ in the matrix $\mathbf{Q}^{-1}$ (here $\mathbf{Q}$ is the matrix of eigenvectors of $(\mathbf{I} + \mathbf{H} \otimes \mathbf{A})$). If multiple eigenvalues have the same dominating magnitude, the final representation $\mathbf{X}_L$ converges to the sum of the dominating terms.*

See the Appendix for a proof. Theorem 2 tells us that whenever a single eigenvalue $(1 + \lambda_j^H \lambda_n^A)$ dominates, *every input in $\mathbf{X}_L$ converges to the same feature vector*. This is because $\mathbf{v}_n^A = \mathbf{1}$ and so $\mathbf{x}_{L,i} \sim \mathbf{v}_j^H$, for all $i$ as $L \to \infty$. On the other hand, whenever $(1 + \lambda_j^H \lambda_1^A)$ singularly dominates, each feature is not guaranteed to be identical. However, in both cases the final representation $\mathbf{X}_L$ is rank one. If instead there are multiple eigenvalue pairs $\lambda_j^H \lambda_i^A$ that equal the value of the dominating eigenvalue, then the rank of $\mathbf{X}_L$ depends on the number of completely unique pairs.

**Corollary 1.** *Let $\mathcal{E}$ be the set of pairs of indices $(j, i)$ such that $|1 + \lambda_j^H \lambda_i^A|$ is equal to the dominating eigenvalue magnitude. Define a unique pair set $\mathcal{U} \subseteq \mathcal{E}$, for which the following holds: $(j, i) \in \mathcal{U}$ iff $(j, i) \in \mathcal{E}$ and $(j, i'), (j', i) \notin \mathcal{U}$, for all $i' \in \{1, \ldots, n\} \setminus i$ and $j' \in \{1, \ldots, d\} \setminus j$. Define a maximal unique pair set $\mathcal{U}^*$ as $|\mathcal{U}^*| \geq |\mathcal{U}|$ for all unique pair sets $\mathcal{U}$. As $L \to \infty$, the rank of $\mathbf{X}_L$ converges to $|\mathcal{U}^*|$.*

The proof is left to the Appendix. Dong et al. (2021) proved that there exist infinitely many parameterizations of $\mathbf{A}$ and $\mathbf{H}$ such that the rank of $\mathbf{X}_L$ does not collapse to 1. For instance, they point out that if $\mathbf{A} = \mathbf{H} = \mathbf{I}$ rank collapse does not occur. Corollary 1 describes exactly which parameterizations do not have rank collapse: only when multiple eigenvalues of $\mathbf{A}$ and $\mathbf{H}$ are perfectly balanced such that multiple eigenvalues simultaneously have equivalent dominant magnitudes, which occurs for $\mathbf{A} = \mathbf{H} = \mathbf{I}$. However, this is a rare case: because $\mathbf{A}$ and $\mathbf{H}$ are learned, it is unlikely the

magnitude of any two eigenvalues $|1 + \lambda_j^H \lambda_i^A|$ will be identical. In these cases, the rank of $\mathbf{X}_L$ will converge to 1, even with the residual connection.

This result on feature convergence allows us to analyze when the Transformer update in eq. (3) is a low-pass filter, and when it is not.

FILTERING

**Theorem 3.** *For all* $\mathbf{X} \in \mathbb{R}^{n \times d}$, *as the number of total layers* $L \to \infty$, *if (1)* $(1 + \lambda_j^H \lambda_n^A)$ *dominates,*

$$\lim_{L \to \infty} \frac{\|\mathrm{HFC}[\mathbf{X}_L]\|_2}{\|\mathrm{LFC}[\mathbf{X}_L]\|_2} = 0, \tag{6}$$

*and so* $(\mathbf{I} + \mathbf{H} \otimes \mathbf{A})$ *acts as a low-pass filter, as in Definition 1. If (2)* $(1 + \lambda_j^H \lambda_1^A)$ *dominates,*

$$\lim_{L \to \infty} \frac{\|\mathrm{HFC}[\mathbf{X}_L]\|_2}{\|\mathrm{LFC}[\mathbf{X}_L]\|_2} \neq 0, \tag{7}$$

*and so* $(\mathbf{I} + \mathbf{H} \otimes \mathbf{A})$ *does not act as a low-pass filter. If (3) multiple eigenvalues have the same dominating magnitude, and there is at least one dominating eigenvalue* $(1 + \lambda_j^H \lambda_i^A)$ *where* $\lambda_i^A \neq \lambda_n^A$, *then eq. 13 holds and* $(\mathbf{I} + \mathbf{H} \otimes \mathbf{A})$ *does not act as a low-pass filter.*

The proof is left to the Appendix. Wang et al. (2022) showed that if we just apply the self-attention matrix $\mathbf{A}$ alone to produce $\mathbf{X}_L$, i.e., $\mathbf{X}_L = \mathbf{A}^L \mathbf{X}$, then this model is always a low-pass filter, as defined in Definition 1. Theorem 3 shows that the residual connection and weights $\mathbf{H}$ can counteract this, so long as condition **(2)** or **(3)** holds.

How can we use these results to improve Transformer models? In the next section we derive a Corollary of Theorem 1 that, along with another observation about the properties $\mathbf{H}$ in trained models, allows us to design a simple reparameterization of $\mathbf{H}$ that guarantees the Transformer update does not oversmooth.

## 4 A REPARAMETERIZATION THAT AVOIDS OVERSMOOTHING

We know from Theorem 2 and 3 that if we can ensure an eigenvalue $(1 + \lambda_j^H \lambda_1^A)$ dominates we avoid oversmoothing. To find $\mathbf{A}$ and $\mathbf{H}$ that are guaranteed to have $(1 + \lambda_j^H \lambda_1^A)$ dominate, we can use Theorem 1. While our results apply to settings where $\mathbf{A}$ and $\mathbf{H}$ are fixed, in practice, both change every layer, and $\mathbf{A}$ changes even every batch $\mathbf{X}$. Because of this, we would like to find a solution that involves only controlling the eigenvalues of $\mathbf{H}$. Luckily, Theorem 1 implies the following much simpler condition.

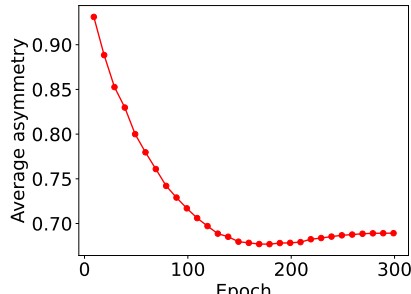

Figure 2: **Asymmetry of the matrix** $\mathbf{H} := \mathbf{W}_{\mathrm{proj}}^\top \mathbf{W}_V^\top$.

**Corollary 2.** *If the eigenvalues of* $\mathbf{H}$ *fall within* $[-1, 0)$, *then at least one of* $\{(1 + \lambda_d^H \lambda_1^A), (1 + \lambda_1^H \lambda_1^A)\}$ *dominates. If the eigenvalues of* $\mathbf{H}$ *fall within* $(0, \infty)$, *then* $(1 + \lambda_d^H \lambda_n^A)$ *dominates.*

See the Appendix for a proof. Corollary 2 states that so long as we can ensure the eigenvalues of $\mathbf{H}$ lie in $[-1, 0)$ we avoid oversmoothing. To ensure that the eigenvalues of $\mathbf{H}$ fall in this range, we propose to directly parameterize its eigendecomposition.

Before doing so, we would like to better understand what $\mathbf{H}$ looks like in a trained model. We made the following observation: as training progresses, $\mathbf{H}$ becomes more symmetric. Specifically, we can measure the asymmetry of any matrix $\mathbf{M}$ as $\frac{\|(\mathbf{M} - \mathbf{M}^\top)/2\|}{\|(\mathbf{M} + \mathbf{M}^\top)/2\|}$, where the numerator measures the asymmetric component of $\mathbf{M}$ and the denominator measures the symmetric component. When we compute this measure for a ViT-Ti on CIFAR100 for all layers, we find that training increases symmetry, as shown in Figure 2. This observation aligns with Trockman & Kolter (2023), who observe that for vision Transformers trained on ImageNet, $\mathbf{H} \propto \epsilon - \mathbf{I}$.

|  | Random Init. | | ImageNet | | CIFAR100 | | | |
|---|---|---|---|---|---|---|---|---|
| **Model** | ViT-B/16 | ViT-Ti* | ViT-B/16 | DeiT-B/16 | ViT-Ti | ViT-Ti$^+$ | ViT-Ti$^-$ | ViT-Ti* |
| $(1 + \lambda_j^H \lambda_n^A)$ | 100% | 50% | 70.27% | 57.82% | 98.92% | 100% | 0% | 50% |
| $(1 + \lambda_j^H \lambda_1^A)$ | 0% | 50% | 29.73% | 42.18% | 1.08% | 0% | 100% | 50% |

Table 1: **Distribution of dominating eigenvalues.** We compare different models trained (or not) on ImageNet and CIFAR100 and count the percentage of cases where the dominating eigenvalue is $(1 + \lambda_j^H \lambda_n^A)$ or $(1 + \lambda_j^H \lambda_1^A)$.

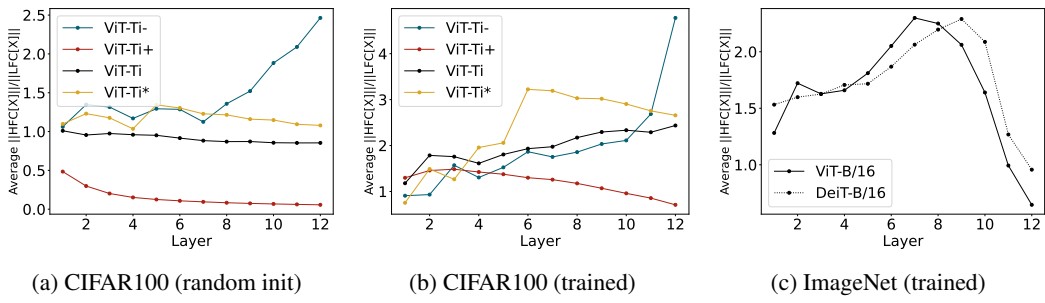

(a) CIFAR100 (random init)    (b) CIFAR100 (trained)    (c) ImageNet (trained)

Figure 3: **Filtering.** $\frac{\|\mathrm{HFC}[\mathbf{X}_\ell]\|_2}{\|\mathrm{LFC}[\mathbf{X}_\ell]\|_2}$ for different models on CIFAR100 and ImageNet.

Given this we propose two different parameterizations of $\mathbf{H}$. The first as a symmetric eigendecomposition $\mathbf{H} = \mathbf{V}_H \Lambda_H \mathbf{V}_H^\top$. A benefit of this is we avoid an expensive and potentially unstable matrix inversion of $\mathbf{V}_H^\top$, required for eigendecomposition of a generic matrix. Instead, $\mathbf{V}_H$ must be an orthogonal matrix. Naively, we could encourage this by adding a term to the loss function of the form $\|\mathbf{V}_H^\top \mathbf{V}_H - \mathbf{I}\|$. However, to guarantee orthogonality we leverage the fact that the QR-decomposition is known to be a differentiable operator Seeger et al. (2017). Therefore we can introduce a parameter matrix $\boldsymbol{\Theta}$ and compute its QR-decomposition to produce $\mathbf{V}_H$, i.e., $[\mathbf{V}_H, \mathbf{R}] := \mathsf{QR}(\boldsymbol{\Theta})$. We then take gradients back to $\boldsymbol{\Theta}$ to learn $\mathbf{V}_H$.

The symmetric parameterization however may restrict the expressiveness of $\mathbf{H}$. Therefore we also test out a non-symmetric eigendecomposition $\mathbf{H} = \mathbf{V}_H \Lambda_H \mathbf{V}_H^{-1}$. In this case, we take gradients to $\mathbf{V}_H$ in the backwards pass, and compute $\mathbf{H}$ using the above equation in the forwards pass.

To ensure $\mathsf{diag}(\Lambda_H) \in [-1, 0)$ given a parameter vector $\boldsymbol{\psi}$ we could naively define it as $\mathsf{diag}(\Lambda_H) = \mathsf{clip}(\boldsymbol{\psi}, [-1, -\epsilon]) \mathbf{V}_H^{-1}$, where $\mathsf{clip}(a, [b, c]) := \min(\max(a, b), c)$ forces all of $\boldsymbol{\psi}$ to lie in $[-1, -\epsilon]$, for arbitrarily small $\epsilon$. However, we noticed that so long as we initialized $\boldsymbol{\psi}$ to fall in $[-1, 0)$, $\mathsf{diag}(\Lambda_H)$ stayed within this range throughout training if we instead just constrained the sign of $\boldsymbol{\psi}$, i.e., $\mathsf{diag}(\Lambda_H) := -(\boldsymbol{\psi}^2)$. For this reason we will refer to our parameterization in the following section using the superscript $^-$. To show a case where oversmoothing is guaranteed, we also define a model $\mathsf{diag}(\Lambda_H) := (\boldsymbol{\psi}^2)$, which we refer to using the superscript $^+$. We also tested a third variant where the first $L/2$ layers sharpen (using the above $^-$ parameterization) and the remaining $L/2$ layers are left normal (i.e., $\mathbf{H}$ is reverted back to $\mathbf{W}_{\mathsf{proj}}^\top \mathbf{W}_V^\top$), which we denote using $^*$. Finally, we label models that use the non-symmetric parameterization of $\mathbf{H}$ with the suffix '-NS'.

## 5 RESULTS

We can now characterize the oversmoothing behavior of different Transformer models. We start by investigating the distribution of dominating eigenvalues and the filtering properties of existing Transformer models and our proposed parameterization. We show that our parameterization, which can be applied to any Transformer model, improves upon the standard parameterization when training data is sparse or corrupted. Crucially, even though our theoretical analysis applies for fixed attention $\mathbf{A}$ and weights $\mathbf{H}$ we will learn different attention and weight matrices for every layer, as is standard practice (further, if a model has multiple heads we will define $\mathbf{W}_V = \mathbf{V}_H$ and $\mathbf{W}_{\mathsf{proj}} = \Lambda_H \mathbf{V}_H^\top$).

**Training and Architecture Details.** For all non-toy experiments we either (a) train variants of the ViT-Ti model Touvron et al. (2021a), or (b) evaluate pretrained Vit-B/16 and DeiT-B/16 Touvron et al. (2021a;a). For (a) we train on CIFAR100 for 300 epochs using the cross-entropy loss and the AdamW optimizer Loshchilov & Hutter (2019). Our setup is the one used in Park & Kim (2022) which itself follows the DeiT training recipe Touvron et al. (2021a). We use a cosine annealing schedule with an initial learning rate of $1.25 \times 10^{-4}$ and weight decay of $5 \times 10^{-2}$. We use a batch size of 96. We use data augmentation including RandAugment Cubuk et al. (2019), CutMix Yun et al. (2019), Mixup Zhang et al. (2018), and label smoothing Touvron et al. (2021a). The models were trained on two Nvidia RTX 2080 Ti GPUs. For (b), ViT-B/16 was pretrained on JFT-300M Sun et al. (2017) before finetuning on ImageNet while DeiT-B/16 was trained only on ImageNet, as its original motivation was to improve data efficiency Touvron et al. (2021a). Importantly, these models do not follow the precise update equation we analyze in our theoretical results: they include layer normalization (LN), arranged in the pre-LN format Xiong et al. (2020). Our goal is to see whether we can still use our theory to control the smoothing/sharpening behavior of such models.

**In popular Transformer models, some H, A do not oversmooth.** We would like to understand to what extent the $\mathbf{H}, \mathbf{A}$ matrices in current models oversmooth. We start by investigating which eigenvalues dominate the Transformer updates of randomly initialized, pre-trained, and newly trained models in Table 1. We obtain $\mathbf{H}$ from the weights $\mathbf{W}_V$ and $\mathbf{W}_{\mathsf{proj}}$ of the pretrained models ViT-B/16 and DeiT-B/16. To compute $\mathbf{A}$ data is needed, on CIFAR100 trained models we use all of the data and on ImageNet trained models we use 50 batches of 8 images making it a total of 400 images. We average over all heads and all layers for $\mathbf{H}$ and $\mathbf{A}$. We notice that the randomly initialized ViT-B/16 has no eigenvalues where $(1 + \lambda_j^H \lambda_1^A)$ is largest, which indicates that all $\mathbf{H}, \mathbf{A}$ will oversmooth. However, once ViT-B/16 is trained (on ImageNet) the percentage of matrices where $(1 + \lambda_j^H \lambda_1^A)$ dominates increases, indicating that there are matrices $\mathbf{H}, \mathbf{A}$ that avoid oversmoothing. It is even larger for the more data efficient DeiT-B/16, but much lower for ViT-Ti on CiFAR100.

**It is possible to guarantee H, A do not oversmooth.** We also compare our reparameterized version of ViT-Ti, which we call ViT-Ti$^-$. Table 1 shows that all of the eigenvalues of $\mathbf{H}, \mathbf{A}$ are of the form $(1 + \lambda_j^H \lambda_1^A)$, confirming Corollary 2. We also compare ViT-Ti$^+$ which is parameterized to have eigenvalues $\lambda_j^H$ that are always positive. As predicted by the first condition in Theorem 1, all eigenvalues are of the form $(1 + \lambda_j^H \lambda_n^A)$.

**Existing models do not converge to low-pass filters.** While the eigenvalue distribution helps us understand the extremes of the eigenvalue distribution, it does not guarantee that a model with a finite depth will cause oversmoothing, by acting as a low-pass filter. To measure how much a model acts as a low-pass filter we compute the quantity described in Definition 1, i.e., $\frac{\|\mathrm{HFC}[\mathbf{X}_\ell]\|_2}{\|\mathrm{LFC}[\mathbf{X}_\ell]\|_2}$ for each layer $\ell$ in existing Transformer models, in Figure 3. We repeat the process done for table 1. For CIFAR100 we measure the quantity using the whole dataset, for ImageNet we randomly sample 50 batches of 8 images making a total of 400 images. We see in that on CIFAR100, the average value of $\frac{\|\mathrm{HFC}[\mathbf{X}_\ell]\|_2}{\|\mathrm{LFC}[\mathbf{X}_\ell]\|_2}$ actually increases for ViT-Ti, indicating that it is in fact suppressing low frequencies instead of amplifying them. Curiously, on ImageNet, both ViT-B/16 and DeiT-B/16 initially suppress low frequencies, but then suppress high frequencies around the 7th layer and deeper: both models dropping $\frac{\|\mathrm{HFC}[\mathbf{X}_\ell]\|_2}{\|\mathrm{LFC}[\mathbf{X}_\ell]\|_2}$ by roughly half the value where each model started. In this case the models begin to smooth the input, which may in fact be useful for classifying certain images (e.g., pooling operations that smooth feature representations can improve classification performance).

**Reparameterization allows one to control Transformer filtering behavior.** Figure 3 also shows the filtering behavior of our reparameterized models ViT-Ti$^-$ and ViT-Ti$^+$. Even though all layers have different $\mathbf{H}, \mathbf{A}$ we see that ViT-Ti$^-$ amplifies higher frequencies, while ViT-Ti$^+$ suppresses them.

**Reparameterization improves robustness to data corruption and data efficiency.** What are the benefits of amplifying higher frequencies using reparameterization? We hypothesize that such a model (ViT-Ti$^-$) will have better performance when (a) training data is limited or (b) test data is corrupted. The intuition for (a) is that amplifying higher frequencies may help the network more

| Training data | 100% | 50% | 10% |
|---|---|---|---|
| ViT-Ti | $66.78_{\pm 0.1}$ | $56.38_{\pm 0.5}$ | $32.47_{\pm 0.9}$ |
| ViT-Ti + FeatScale | $66.57_{\pm 0.5}$ | $57.08_{\pm 0.3}$ | $32.50_{\pm 0.4}$ |
| ViT-Ti$^+$ | $64.34_{\pm 0.3}$ | $55.21_{\pm 0.5}$ | $28.67_{\pm 0.5}$ |
| ViT-Ti$^-$ | $66.62_{\pm 0.3}$ | $56.58_{\pm 0.3}$ | $\mathbf{33.99_{\pm 0.4}}$ |
| ViT-Ti$^*$ | $67.84_{\pm 0.6}$ | $\mathbf{57.78_{\pm 0.4}}$ | $33.64_{\pm 0.5}$ |
| ViT-Ti-NS$^-$ | $67.80_{\pm 0.2}$ | $57.54_{\pm 0.1}$ | $\mathbf{33.78_{\pm 0.3}}$ |
| ViT-Ti-NS$^*$ | $\mathbf{68.53_{\pm 0.6}}$ | $\mathbf{57.67_{\pm 0.3}}$ | $33.75_{\pm 0.2}$ |

| Depth | 12 | 24 |
|---|---|---|
| ViT-Ti | $66.78_{\pm 0.1}$ | $67.22_{\pm 0.1}$ |
| ViT-Ti + FeatScale | $66.57_{\pm 0.5}$ | $67.71_{\pm 0.5}$ |
| ViT-Ti$^+$ | $64.34_{\pm 0.3}$ | $65.65_{\pm 0.3}$ |
| ViT-Ti$^-$ | $66.62_{\pm 0.3}$ | $66.76_{\pm 0.3}$ |
| ViT-Ti$^*$ | $67.84_{\pm 0.6}$ | $69.05_{\pm 0.1}$ |
| ViT-Ti-NS$^-$ | $67.80_{\pm 0.2}$ | $67.09_{\pm 0.1}$ |
| ViT-Ti-NS$^*$ | $\mathbf{68.53_{\pm 0.6}}$ | $\mathbf{69.64_{\pm 0.3}}$ |

Table 2: **Data Efficiency.** Test accuracy when training on different amounts of the CIFAR100 training set.

Table 3: **Increased Depth.** Test accuracy for different model depths on CIFAR100.

| Corruption Intensity | 0 | 1 | 2 | 3 | 4 | 5 |
|---|---|---|---|---|---|---|
| ViT-Ti | $66.78_{\pm 0.1}$ | $61.59_{\pm 0.2}$ | $57.75_{\pm 0.2}$ | $53.12_{\pm 0.1}$ | $49.32_{\pm 0.1}$ | $39.17_{\pm 0.2}$ |
| ViT-Ti + Featscale | $66.57_{\pm 0.5}$ | $61.25_{\pm 0.5}$ | $57.36_{\pm 0.6}$ | $52.64_{\pm 0.6}$ | $48.87_{\pm 0.5}$ | $38.77_{\pm 0.4}$ |
| ViT-Ti$^+$ | $64.34_{\pm 0.3}$ | $58.36_{\pm 0.4}$ | $53.83_{\pm 0.5}$ | $49.03_{\pm 0.6}$ | $45.15_{\pm 0.6}$ | $35.86_{\pm 0.6}$ |
| ViT-Ti$^-$ | $66.62_{\pm 0.3}$ | $61.62_{\pm 0.2}$ | $58.00_{\pm 0.3}$ | $53.56_{\pm 0.2}$ | $49.70_{\pm 0.3}$ | $40.07_{\pm 0.2}$ |
| ViT-Ti$^*$ | $67.84_{\pm 0.6}$ | $62.49_{\pm 0.5}$ | $58.82_{\pm 0.7}$ | $54.32_{\pm 0.7}$ | $50.40_{\pm 0.6}$ | $40.38_{\pm 0.7}$ |
| ViT-Ti-NS$^-$ | $67.80_{\pm 0.2}$ | $\mathbf{63.03_{\pm 0.2}}$ | $\mathbf{59.65_{\pm 0.1}}$ | $\mathbf{55.05_{\pm 0.3}}$ | $\mathbf{51.30_{\pm 0.1}}$ | $\mathbf{41.33_{\pm 0.1}}$ |
| ViT-Ti-NS$^*$ | $\mathbf{68.53_{\pm 0.6}}$ | $63.01_{\pm 0.1}$ | $59.44_{\pm 0.3}$ | $54.90_{\pm 0.4}$ | $51.12_{\pm 0.4}$ | $40.90_{\pm 0.4}$ |

Table 4: **Corruption Robustness.** Test accuracy for different corruption intensities on CIFAR100.

easily identify key features that distinguish different classes, while smoothing may make this more difficult. For (b), amplifying higher frequencies may help to preserve information that is useful for classification, even if some that information is removed by corruption. Whereas, a low-pass filter may remove any identifying information that is left.

To test these hypotheses, we run two sets of experiments: 1. **Data efficiency**: we train ViT-Ti, ViT-Ti$^-$, ViT-Ti$^+$ on fractions of CIFAR100 including $\{10\%, 50\%, 100\%\}$; 2. **Corruption robustness**: we take ViT-Ti, ViT-Ti$^-$, ViT-Ti$^+$ trained on all of CIFAR100 and we test it with 15 types of corruptions as described in Hendrycks & Dietterich (2019), at 5 different levels of intensity.

Table 2 reports the results of the data efficiency experiment. We find that ViT-Ti$^-$ performs best in lower data settings while ViT-Ti$^+$ performs worst, confirming our intuition. On the original CIFAR100 ViT-Ti performs best, we suspect this is because it has been designed to perform well on this dataset. Table 4 shows the results of the corruption robustness experiment. ViT-Ti$^-$ outperforms ViT and ViT-Ti$^+$ on all corruption intensities, and ViT-Ti$^+$ always performs worst. Again on the original clean data ViT-Ti is best.

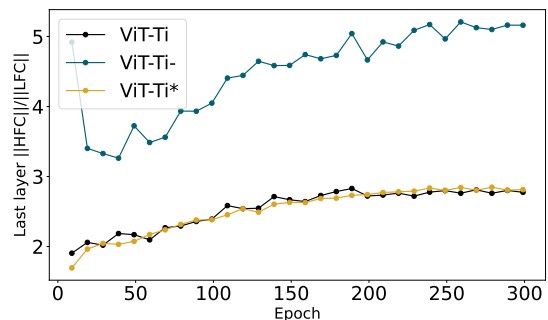

**Pushing performance further by avoiding oversharpening** While the parameterization used for ViT-Ti$^-$ helps guarantee that we avoid oversmoothing, it loses the flexible ability unconstrained layers have to learn both high-pass and low-pass filters. The model only having

Figure 4: **Evolution of the last layer** $\frac{\|\mathrm{HFC}[\mathbf{X}_\ell]\|_2}{\|\mathrm{LFC}[\mathbf{X}_\ell]\|_2}$ The ratio for different models on CIFAR100 during training.

sharpening layers can lead to an oversharpening of the features which also affects performance. The accuracy of the ViT-TI$^-$ model barely improves after we double the depth. ViT-Ti$^*$ avoids this issue due to having unconstrained layers. It significantly outperforms all other models not just for larger depths in Table 3, but also in corruption robustness (Table 4), and data efficiency (Table 2). This suggests that both oversmoothing and oversharpening are detrimental to performance, and that a careful balance is key to an accurate model. We can see from Fig. 4 that ViT-Ti$^-$ has a much higher last layer low-pass filtering ratio $\frac{\|\mathrm{HFC}[\mathbf{X}_\ell]\|_2}{\|\mathrm{LFC}[\mathbf{X}_\ell]\|_2}$ while ViT-Ti$^*$ follows ViT-Ti. Fig. 3 however shows

that ViT-Ti* initially sharpens the features before smoothing them, while ViT-Ti slightly sharpens throughout the network.

## 6 RELATED WORK

Oversmoothing is a concept that has been widely discussed in the graph neural network literature Rusch et al. (2023). Giovanni et al. (2023) prove that graph convolutions can enhance high frequencies and have an oversharpening asymptotic behaviour. Their analysis inspires our work. For transformers, Zhou et al. (2021) noticed that as depth was increased, the cosine similarity between self-attention matrices $\mathbf{A}$ also increased. Similarly, Gong et al. (2021) and Raghu et al. (2021) found that feature similarity in vision Transformers increased with depth. Many works around this time found that it was possible to improve vision Transformers by replacing self-attention layers with convolutional layers (Han et al., 2021; Liu et al., 2021b; Jiang et al., 2021; Touvron et al., 2021b; Yuan et al., 2021; Park & Kim, 2022). Other works introduced new layers to avoid oversmoothing (Wang et al., 2022; Guo et al., 2023; Ali et al., 2023). Oversmoothing has also found to occur in Transformer architectures for NLP Shi et al. (2022). We focus our experimental analysis on vision Transformers as the problem has received greater attention in computer vision, but our analysis applies to any model that uses Transformer blocks.

The first works to develop a theory around Transformer oversmoothing were Dong et al. (2021); Wang et al. (2022). Initially, Dong et al. (2021) showed that, without skip-connections, repeated self-attention layers converge doubly exponentially to a rank 1 matrix. They show that there exist models where skip-connections counteract this convergence. We extend this result to show that this only occurs when multiple unique pairs of eigenvalues all have the same dominating magnitude (Corollary 1). Wang et al. (2022) analyzed oversmoothing from the lens of signal processing. They showed that as the number of self-attention operations tended to infinity, all inputs converge to the same feature vector, producing a low-pass filter. They also analyzed the convergence rate when the residual connection, weights, multiple heads, and a linear layer is added, and found that convergence is not guaranteed. However, they still argued that even with these additions: 'it is inevitable that high-frequency components are continuously diluted as ViT goes deeper'. Our analysis shows that this is not necessarily the case: the addition of the residual connection and weights $\mathbf{H}$, can actually cause high-frequency components to increase as layers are added, as is shown in Figure 2 (a) and (b).

## 7 DISCUSSION

In this paper, we presented a new analysis detailing how the eigenspectrum of attention and weight matrices impacts the final representation produced by the Transformer update, as depth is increased. Contrary to prior work, this analysis revealed that Transformers are not inherently low-pass filters. Unexpectedly however, it also shows that whenever a single eigenvalue of the update equations dominates (which is most likely) the final representation will have rank 1, even with the residual connection. Empirically we show that existing Transformer models already have properties that partially prevent oversmoothing. Finally we introduce a new parameterization for the Transformer weights that is guaranteed to avoid oversmoothing. This parameterization can be used with any Transformer architecture, and it improves ViT-Tiny performance when training data is limited or corrupted, and when the model is made deeper.

**Limitations and Future Work.** One limitation of the current theoretical analysis is that the results are asymptotic, applying in the limit as $L \to \infty$. However, we did find that adapting the eigenspectrum of the model to agree with the analysis empirically improved data efficiency, corruption, and training deeper layers, for a finite number of layers. This said, it would be useful to understand the rates of convergence of each of the quantities in Theorems 2 and 3. Another is the missing analyses of multi-head attention as well as the linear layer that usually follows the residual connection. We suspect a similar analysis for multi-head attention to be difficult as it would require relating the eigenvalues of each individual head $h$, i.e., $(\mathbf{H}_h \otimes \mathbf{A}_h)$ to the eigenvalues of the sum of head outputs $\sum_h (\mathbf{H}_h \otimes \mathbf{A}_h)$. Special conditions would need to be placed on $\mathbf{H}_h, \mathbf{A}_h$ (e.g., that all $(\mathbf{H}_h \otimes \mathbf{A}_h)$ have the same set of eigenvectors). On the other hand including a linear layer after the residual connection in the analysis would likely be straightforward: it could be absorbed into $\mathbf{H}$ and would form a Kronecker product with the $\mathbf{I}$ term in eq. 3. We leave these extensions for future work.

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

## APPENDIX

## A    PROOFS

**Proposition 1** (Meyer & Stewart (2023)). *Given Assumption 1, all eigenvalues of $\mathbf{A}$ lie within $(-1, 1]$. There is one largest eigenvalue that is equal to $1$, with corresponding unique eigenvector $\mathbf{1}$. No eigenvectors of $\mathbf{A}$ are equal to $0$.*

*Proof.* First, because $\mathbf{A}$ is positive, by the Perron-Frobenius Theorem Meyer & Stewart (2023) all eigenvalues of $\mathbf{A}$ are in $\mathbb{R}$ (and so there exist associated eigenvectors that are also in $\mathbb{R}$). Next, recall the definition of an eigenvalue $\lambda$ and eigenvector $\mathbf{v}$: $\mathbf{Av} = \lambda\mathbf{v}$. Let us write the equation for any row $i \in \{1, \ldots, n\}$ explicitly:

$$a_{i1}v_1 + \cdots + a_{in}v_n = \lambda v_i.$$

Further let,

$$v_{\max} := \max\{|v_1|, \ldots, |v_n|\} \tag{8}$$

Note that $v_{\max} > 0$, otherwise it is not a valid eigenvector. Further let $k_{\max}$ be the index of $\mathbf{v}$ corresponding to $v_{\max}$. Then we have,

$$\begin{aligned}
|\lambda|v_{\max} &= |a_{k_{\max}1}v_1 + \cdots + a_{k_{\max}n}v_n| \\
&\leq a_{k_{\max}1}|v_1| + \cdots + a_{k_{\max}n}|v_n| \\
&\leq a_{k_{\max}1}|v_{k_{\max}}| + \cdots + a_{k_{\max}n}|v_{k_{\max}}| \\
&= (a_{k_{\max}1} + \cdots + a_{k_{\max}n})|v_{k_{\max}}| = |v_{\max}|
\end{aligned}$$

The first inequality is given by the triangle inequality and because $a_{ij} > 0$. The second is given by the definition of $v_{\max}$ as the maximal element in $\mathbf{v}$. The final inequality is given by the definition of $\mathbf{A}$ in eq. (1) as right stochastic (i.e., all rows of $\mathbf{A}$ sum to 1) and because $|v_{k_{\max}}| = |v_{\max}|$. Next, note that because $v_{\max} > 0$, it must be that $\lambda \leq 1$. Finally, to show that the one largest eigenvalue is equal to 1, recall by the definition of $\mathbf{A}$ in eq. (1) that $\mathbf{A1} = \mathbf{1}$, where $\mathbf{1}$ is the vector of all ones. So $\mathbf{1}$ is an eigenvector of $\mathbf{A}$, with eigenvalue $\lambda^* = 1$. Because $a_{ij} > 0$, and we showed above that all eigenvalues must lie in in $[-1, 1]$, by the Perron-Frobenius theorem Meyer & Stewart (2023) $\lambda^* = 1$ is the Perron root. This means that all other eigenvalues $\lambda_i$ satisfy the following inequality $|\lambda_i| < \lambda^*$. Further $\mathbf{1}$ is the Perron eigenvector, and all other eigenvectors have at least one negative component, making $\mathbf{1}$ unique. Finally, because $\mathbf{A}$ is invertible, it cannot have any $0$ eigenvalues Brualdi & Mellendorf (1994). □

**Theorem 1.** *Consider the Transformer update with fixed $\mathbf{A} > 0, \mathbf{H} := \mathbf{W}_{\mathsf{proj}}^\top \mathbf{W}_V^\top$, as described in eq. (3). Let $\{\lambda_i^A, \mathbf{v}_i^A\}_{i=1}^n$ and $\{\lambda_j^H, \mathbf{v}_j^H\}_{j=1}^d$ be the eigenvalue and eigenvectors of $\mathbf{A}$ and $\mathbf{H}$. Let the eigenvalues (and associated eigenvectors) be sorted as follows, $\lambda_1^A \leq \cdots \leq \lambda_n^A$ and*

$|1 + \lambda_1^H| \leq \cdots \leq |1 + \lambda_d^H|$. *Let $\varphi_1^H, \ldots, \varphi_d^H$ be the phases of $\lambda_1^H, \ldots, \lambda_d^H$. As the number of layers $L \to \infty$, one eigenvalue dominates the rest (multiple dominate if there are ties):*

$$
\begin{cases}
\begin{rcases}
(1 + \lambda_d^H \lambda_n^A) & \text{if } |1 + \lambda_d^H \lambda_n^A| > |1 + \lambda_d^H \lambda_1^A| \\
(1 + \lambda_1^H \lambda_1^A) & \text{if } |1 + \lambda_d^H \lambda_n^A| < |1 + \lambda_d^H \lambda_1^A|
\end{rcases} & \text{if } \lambda_1^A > 0 \\[2ex]
\begin{rcases}
(1 + \lambda_d^H \lambda_n^A) & \text{if } |1 + \lambda_d^H \lambda_n^A| > |1 + \lambda_k^H \lambda_1^A| \\
(1 + \lambda_k^H \lambda_1^A) & \text{if } |1 + \lambda_d^H \lambda_n^A| < |1 + \lambda_k^H \lambda_1^A|
\end{rcases} & \text{if } \lambda_1^A < 0, \varphi_d^H \in [-\frac{\pi}{2}, \frac{\pi}{2}] \\[2ex]
\begin{rcases}
(1 + \lambda_d^H \lambda_n^A) & \text{if } |1 + \lambda_d^H \lambda_n^A| > |1 + \lambda_d^H \lambda_1^A| \\
(1 + \lambda_d^H \lambda_1^A) & \text{if } |1 + \lambda_d^H \lambda_n^A| < |1 + \lambda_d^H \lambda_1^A|
\end{rcases} & \text{if } \lambda_1^A < 0, \varphi_d^H \in (\frac{\pi}{2}, \pi] \cup [-\pi, -\frac{\pi}{2})
\end{cases}
$$

*where $\lambda_k^H$ is the eigenvalue with the largest index $k$ such that $\varphi_k^H \in (\pi/2, \pi] \cup [-\pi, -\pi/2)$.*

*Proof.* First, note that given the eigendecompositions of $\mathbf{H} := \mathbf{W}_{\text{proj}}^\top \mathbf{W}_V^\top$ and $\mathbf{A} > 0$, as $\{\lambda_i^A, \mathbf{v}_i^A\}_{i=1}^n$ and $\{\lambda_j^H, \mathbf{v}_j^H\}_{j=1}^d$, the eigenvalues and eigenvectors of $(\mathbf{I} + \mathbf{H} \otimes \mathbf{A})$ are equal to $(1 + \lambda_j^H \lambda_i^A)$ and $\mathbf{v}_j^H \otimes \mathbf{v}_i^A$ for all $j \in \{1, ..., d\}$ and $i \in \{1, \ldots, n\}$ (Schacke, 2004, Theorem 2.3). Recall that eigenvalues (and associated eigenvectors) are sorted in the following order $\lambda_1^A \leq \cdots \leq \lambda_n^A$ and $|1 + \lambda_1^H| \leq \cdots \leq |1 + \lambda_d^H|$. Now at least one of the eigenvalues of $(\mathbf{I} + \mathbf{H} \otimes \mathbf{A})$, i.e., $(1 + \lambda_j^H \lambda_i^A)$ has a larger magnitude than all others, i.e., there exists $j^*, i^*$ (which may be a set of indices if there are ties) such that $|1 + \lambda_{j^*}^H \lambda_{i^*}^A| > |1 + \lambda_{j'}^H \lambda_{i'}^A|$ for all $j' \in \{1, \ldots, d\} \setminus j^*$ and $i' \in \{1, \ldots, n\} \setminus i^*$. As $L \to \infty$ the expression $(1 + \lambda_{j^*}^H \lambda_{i^*}^A)^L$ will dominate all eigenvalue expressions $(1 + \lambda_{j'}^H \lambda_{i'}^A)^L$ (again multiple will if there are ties). Our goal is to understand the identity of $\lambda_{j^*}^H \lambda_{i^*}^A$ for all possible values of $\lambda_H, \lambda_A$. For simplicity we will assume there are no ties, i.e., $j^*, i^*$ each denote a single index. In this case we only need to consider strict inequalities of $\lambda_H, \lambda_A$ (as equalities indicate that multiple eigenvalues dominate).

First recall that $\lambda_i^A \in (-1, 1]$ and $\lambda_n^A = 1$. A useful way to view selecting $\lambda_j^H \lambda_i^A$ to maximize $|1 + \lambda_j^H \lambda_i^A|$ is as maximizing distance to $-1$. If (i), $\lambda_1^A > 0$ then $\lambda_1^A$ always shrinks $\lambda_j^H$ to the origin. If $\varphi_j^H \in [-\pi/2, \pi/2]$ then this shrinking will always bring $\lambda_j^H$ closer to $-1$. If instead $\varphi_j^H \in (\pi/2, \pi] \cup [-\pi, -\pi/2)$ then this shrinking can bring $\lambda_j^H$ farther from $-1$. The eigenvalue it can bring farthest from $-1$ is $\lambda_1^H$ (as $\lambda_1^H$ is already farthest from $-1$ given that $|1 + \lambda_1^H| \leq \cdots \leq |1 + \lambda_d^H|$). If this point is farther from $-1$ than $\lambda_d^H \lambda_n^A$, i.e., if $|1 + \lambda_1^H \lambda_1^A| > |1 + \lambda_d^H \lambda_n^A|$ then $(1 + \lambda_1^H \lambda_1^A)$ dominates. Otherwise, $(1 + \lambda_d^H \lambda_n^A)$ dominates. If instead (ii), $\lambda_1^A < 0$ then it is possible to 'flip' $\lambda_j^H$ across the origin, and so the maximizer depends on $\varphi_d^H$. If a) $\varphi_d^H \in [-\pi/2, \pi/2]$ then let $\lambda_k^H$ be the eigenvalue with the largest index $k$ such that $\varphi_k^H \in (\pi/2, \pi] \cup [-\pi, -\pi/2)$. It is possible that 'flipping' this eigenvalue across the origin makes it farther away than $\lambda_d^H$, i.e., $|1 + \lambda_k^H \lambda_1^A| > |1 + \lambda_d^H \lambda_n^A|$. In this case $(1 + \lambda_k^H \lambda_1^A)$ dominates, otherwise $(1 + \lambda_d^H \lambda_n^A)$ dominates. If instead b) $\varphi_d^H \in (\pi/2, \pi] \cup [-\pi, -\pi/2)$ then either $|1 + \lambda_d^H \lambda_n^A| > |1 + \lambda_{j'}^H \lambda_{i'}^A|$ for all $j' \neq d$ and $i' \neq n$, and so $(1 + \lambda_d^H \lambda_n^A)$ dominates, or 'flipping' $\lambda_d^H$ increases its distance from $-1$, and so $|1 + \lambda_d^H \lambda_1^A| > |1 + \lambda_{j'}^H \lambda_{i'}^A|$ for all $j' \neq d$ and $i' \neq n$, and so $(1 + \lambda_d^H \lambda_1^A)$ dominates. $\qquad\square$

**Theorem 2.** *As the number of total layers $L \to \infty$, the feature representation $\mathbf{X}_L$ converges. Which representation it converges to depends on the dominating eigenvalue, as given in Theorem 1. If a single eigenvalue dominates, there are two cases: (1) If $(1 + \lambda_j^H \lambda_n^A)$ dominates then,*

$$
\mathbf{X}_L \to (1 + \lambda_j^H \lambda_n^A)^L s_{j,n} \mathbf{1} \mathbf{v}_j^{H^\top}, \tag{9}
$$

*(2) If $(1 + \lambda_j^H \lambda_1^A)$ dominates then,*

$$
\mathbf{X}_L \to (1 + \lambda_j^H \lambda_1^A)^L s_{j,1} \mathbf{v}_1^A \mathbf{v}_j^{H^\top} \tag{10}
$$

*where $s_{j,i} := \langle \mathbf{v}_{j,i}^{Q^{-1}}, \text{vec}(\mathbf{X}) \rangle$ and $\mathbf{v}_{j,i}^{Q^{-1}}$ is row $ji$ in the matrix $\mathbf{Q}^{-1}$ (here $\mathbf{Q}$ is the matrix of eigenvectors of $(\mathbf{I} + \mathbf{H} \otimes \mathbf{A})$). If multiple eigenvalues have the same dominating magnitude, the final representation $\mathbf{X}_L$ converges to the sum of the dominating terms.*

*Proof.* Recall that the eigenvalues and eigenvectors of $(\mathbf{I} + \mathbf{H} \otimes \mathbf{A})$ are equal to $(1 + \lambda_j^H \lambda_i^A)$ and $\mathbf{v}_j^H \otimes \mathbf{v}_i^A$ for all $j \in \{1, ..., d\}$ and $i \in \{1, \ldots, n\}$. This means,

$$\text{vec}(\mathbf{X}_L) = \sum_{i,j} (1 + \lambda_j^H \lambda_i^A)^L \langle \mathbf{v}^{Q^{-1}}_{j,i}, \text{vec}(\mathbf{X}) \rangle (\mathbf{v}_j^H \otimes \mathbf{v}_i^A). \tag{11}$$

Recall that $\mathbf{v}^{Q^{-1}}_{j,i}$ is row $ji$ in the matrix $\mathbf{Q}^{-1}$, where $\mathbf{Q}$ is the matrix of eigenvectors $\mathbf{v}_j^H \otimes \mathbf{v}_i^A$. Further recall that $\mathbf{v}_i^A = \mathbf{1}$. As described in Theorem 1, as $L \to \infty$ at least one of the eigenvalues pairs $\lambda_j^H \lambda_i^A$ will dominate the expression $(1 + \lambda_j^H \lambda_i^A)^L$, which causes $\text{vec}(\mathbf{X}_L)$ to converge to the dominating term. Finally, we can rewrite, $\mathbf{v}_1 \otimes \mathbf{v}_2$ as $\text{vec}(\mathbf{v}_2 \mathbf{v}_1^\top)$. Now all non-scalar terms have $\text{vec}(\cdot)$ applied, so we can remove this function everywhere to give the matrix form given in eq. (9) and eq. (10). $\square$

**Corollary 1.** *Let $\mathcal{E}$ be the set of pairs of indices $(j, i)$ such that $|1 + \lambda_j^H \lambda_i^A|$ is equal to the dominating eigenvalue magnitude. Define a* unique pair set *$\mathcal{U} \subseteq \mathcal{E}$, for which the following holds: $(j, i) \in \mathcal{U}$ iff $(j, i) \in \mathcal{E}$ and $(j, i'), (j', i) \notin \mathcal{U}$, for all $i' \in \{1, \ldots, n\} \setminus i$ and $j' \in \{1, \ldots, d\} \setminus j$. Define a maximal* unique pair set *$\mathcal{U}^*$ as $|\mathcal{U}^*| \geq |\mathcal{U}|$ for all unique pair sets $\mathcal{U}$. As $L \to \infty$, the rank of $\mathbf{X}_L$ converges to $|\mathcal{U}^*|$.*

*Proof.* First recall that the rank of a matrix $\mathbf{M}$ is the smallest number $k$ such that $\mathbf{M}$ can be written as a sum of $k$ rank-1 matrices. Next note that if we have $\mathcal{E} = \{(j, i), (j', i)\}$ for $j \neq j'$ then we have

$$\mathbf{X}_L \to a_{j,i} \mathbf{v}_i^A \mathbf{v}_j^{H\top} + a_{j',i} \mathbf{v}_i^A \mathbf{v}_{j'}^{H\top} = a_{j,i} \mathbf{v}_i^A (\mathbf{v}_j^{H\top} + \frac{a_{j',i}}{a_{j,i}} \mathbf{v}_{j'}^{H\top}),$$

where $a_{j,i} := (1 + \lambda_j^H \lambda_i^A)^L s_{j,i}$. This shows that $\mathbf{X}_L$ is rank 1, which agrees in this example with $|\mathcal{U}^*| = 1$ (the same holds for $\mathcal{E} = \{(j, i), (j, i')\}$). In general, whenever the same index appears in different pairs in $\mathcal{E}$, we can group all associated terms in the expression for $\mathbf{X}_L$ into a single rank-1 term. Therefore, an element in a unique pair set $\mathcal{U}$ corresponds to a grouped rank-1 term in the expression for $\mathbf{X}_L$. Every element in a maximal unique pair set $\mathcal{U}^*$ corresponds 1-to-1 to every grouped rank-1 term in the expression for $\mathbf{X}_L$. So we can write $\mathbf{X}_L$ as,

$$\mathbf{X}_L \to \sum_{(j,i) \in \mathcal{U}^*} a_{j,i} \mathbf{g}_i^A \mathbf{g}_j^{H\top}, \tag{12}$$

where each $\mathbf{g}_i^A, \mathbf{g}_j^H$ are potentially grouped terms (i.e., linear combinations of $\mathbf{v}_i^A, \mathbf{v}_j^H$). Further, none of the elements of the above sum can be grouped to yield a sum with fewer rank-1 terms. Therefore, the rank of $\mathbf{X}_L$ approaches $|\mathcal{U}^*|$, and we are done. $\square$

**Corollary 2.** *If the eigenvalues of $\mathbf{H}$ fall within $[-1, 0)$, then at least one of $\{(1 + \lambda_d^H \lambda_1^A), (1 + \lambda_1^H \lambda_1^A)\}$ dominates. If the eigenvalues of $\mathbf{H}$ fall within $(0, \infty)$, then $(1 + \lambda_d^H \lambda_n^A)$ dominates.*

*Proof.* Let $\lambda_1^H \leq \cdots \leq \lambda_d^H$. Again we can think of selecting $\lambda_j^H \lambda_i^A$ that maximizes $|1 + \lambda_j^H \lambda_i^A|$ as maximizing the distance of $\lambda_j^H \lambda_i^A$ to $-1$. Consider the first case where $\lambda_1^H, \cdots, \lambda_d^H \in [-1, 0)$, and so $\lambda_1^H$ is the closest eigenvalue to $-1$ and $\lambda_d^H$ is the farthest. If $\lambda_1^A > 0$ then all $\lambda^A$ can do is shrink $\lambda^H$ to the origin, where $\lambda_1^A$ shrinks $\lambda^H$ the most. The closest eigenvalue to the origin is $\lambda_d^H$, and so $(1 + \lambda_d^H \lambda_1^A)$ dominates. If instead $\lambda_1^A < 0$, then we can 'flip' $\lambda_j^H$ over the origin, making it farther from $-1$ than all other $\lambda_{j'}^H$. The eigenvalue that we can 'flip' the farthest from $-1$ is $\lambda_1^H$, and so $(1 + \lambda_1^H \lambda_1^A)$ dominates. If all eigenvalues of $\mathbf{H}$ are equal, then both $(1 + \lambda_d^H \lambda_1^A)$ and $(1 + \lambda_1^H \lambda_1^A)$ dominate. For the second case where $\lambda_1^H, \cdots, \lambda_d^H \in (0, \infty)$ the result follows directly from the first case in Theorem 4, and so we are done. $\square$

**Corollary 3.** *If the phases of $\lambda_1^H, \ldots, \lambda_d^H$ all fall within specific ranges, the dominating eigenvalue conditions can be simplified as follows:*

$$
\begin{cases}
(1 + \lambda_d^H \lambda_n^A) & & \text{if } \varphi_j^H \in [-\frac{\pi}{2}, \frac{\pi}{2}], \forall j \\
\left.\begin{array}{ll}
(1 + \lambda_d^H \lambda_n^A) & \text{if } |1 + \lambda_d^H \lambda_n^A| > |1 + \lambda_1^H \lambda_1^A| \\
(1 + \lambda_1^H \lambda_1^A) & \text{if } |1 + \lambda_d^H \lambda_n^A| < |1 + \lambda_1^H \lambda_1^A|
\end{array}\right\} & \text{if } \lambda_1^A > 0 \\
\left.\begin{array}{ll}
(1 + \lambda_d^H \lambda_n^A) & \text{if } |1 + \lambda_d^H \lambda_n^A| > |1 + \lambda_d^H \lambda_1^A| \\
(1 + \lambda_d^H \lambda_1^A) & \text{if } |1 + \lambda_d^H \lambda_n^A| < |1 + \lambda_d^H \lambda_1^A|
\end{array}\right\} & \text{if } \lambda_1^A < 0
\end{cases}
\quad \text{if } \varphi_j^H \in (\frac{\pi}{2}, \pi] \cup [-\pi, -\frac{\pi}{2}), \forall j
$$

*Proof.* The proof is similar to that of Theorem 1 except here we consider special cases.

**if $\varphi_j^H \in [-\pi/2, \pi/2]$ for all $j \in \{1, \ldots, d\}$.** First recall that $\lambda_i^A \in (-1, 1]$ and $\lambda_n^A = 1$. As $\varphi_j^H \in [-\pi/2, \pi/2]$, we have that $|1 + \lambda_d^H \lambda_n^A| > |1 + \lambda_{j'}^H \lambda_{i'}^A|$ for all $j' \in \{1, \ldots, d-1\}$ and $i' \in \{1, \ldots, n-1\}$. This is because, if (i), $\lambda_1^A > 0$, (and so $\lambda_i^A > 0$ for all $i$) then $\arg(\lambda_{j'}^H \lambda_{i'}^A) \in [-\pi/2, \pi/2]$, where $\arg(a)$ is the argument or 'phase' or $a \in \mathbb{C}$. This combined with the fact that $|\lambda_{j'}^H \lambda_{i'}^A| < |\lambda_d^H \lambda_n^A|$ means that $|1 + \lambda_{j'}^H \lambda_{i'}^A| < |1 + \lambda_d^H \lambda_n^A|$. This because for any two points $a, a'$ where we have that $\arg(a), \arg(a') \in [-\pi/2, \pi/2]$ and $|a'| < |a|$, then it also holds that $|1 + a'| < |1 + a|$. Therefore $(1 + \lambda_d^H \lambda_n^A)$ dominates. If instead (ii), $\lambda_1^A < 0$ then for any negative eigenvalues $\lambda_{i-}^A < 0$ we have that $\arg(\lambda_j^H \lambda_{i-}^A) \in (\pi/2, \pi] \cup [-\pi, -\pi/2)$ for all $j$. However, for each of these points we have that $|1 + \lambda_j^H \lambda_{i-}^A| < |1 + \lambda_j^H \lambda_n^A|$. This is because for any $r \in (-1, 0)$ and point $b$ where $\arg(b) \in [-\pi/2, \pi/2]$ we have that $|1 + r * b| < |1 + b|$. Further note that $|1 + \lambda_j^H \lambda_n^A| < |1 + \lambda_d^H \lambda_n^A|$ from our definitions: $\lambda_n^A = 1$ and $|1 + \lambda_1^H| < \cdots < |1 + \lambda_d^H|$. And so $(1 + \lambda_d^H \lambda_n^A)$ dominates. For the remaining positive eigenvalues $\lambda_{i+}^A \geq 0$ we are in the same situation as (i), and so we are done.

**if $\varphi_j^H \in (\pi/2, \pi] \cup [-\pi, -\pi/2)$ for all $j \in \{1, \ldots, d\}$.** If (a), $\lambda_1^A > 0$ then either $|1 + \lambda_d^H \lambda_n^A| > |1 + \lambda_{j'}^H \lambda_{i'}^A|$ for all $j' \neq d$ and $i' \neq n$, and so $(1 + \lambda_d^H \lambda_n^A)$ dominates, or shrinking $\lambda_1^H$ to the origin makes it the farthest from $-1$, i.e, $|1 + \lambda_1^H \lambda_1^A| > |1 + \lambda_{j'}^H \lambda_{i'}^A|$, and so $(1 + \lambda_1^H \lambda_1^A)$ dominates. If (b) $\lambda_1^A < 0$ then either $|1 + \lambda_d^H \lambda_n^A| > |1 + \lambda_{j'}^H \lambda_{i'}^A|$, and so $(1 + \lambda_d^H \lambda_n^A)$ dominates, or 'flipping' $\lambda_d^H$ across the origin makes it farthest from $-1$, i.e., $|1 + \lambda_d^H \lambda_1^A| > |1 + \lambda_{j'}^H \lambda_{i'}^A|$, and so $(1 + \lambda_d^H \lambda_1^A)$ dominates. $\square$

**Theorem 3.** *For all $\mathbf{X} \in \mathbb{R}^{n \times d}$, as the number of total layers $L \to \infty$, if (1) $(1 + \lambda_j^H \lambda_n^A)$ dominates,*

$$\lim_{L \to \infty} \frac{\|\text{HFC}[\mathbf{X}_L]\|_2}{\|\text{LFC}[\mathbf{X}_L]\|_2} = 0, \tag{13}$$

*and so $(\mathbf{I} + \mathbf{H} \otimes \mathbf{A})$ acts as a low-pass filter, as in Definition 1. If (2) $(1 + \lambda_j^H \lambda_1^A)$ dominates,*

$$\lim_{L \to \infty} \frac{\|\text{HFC}[\mathbf{X}_L]\|_2}{\|\text{LFC}[\mathbf{X}_L]\|_2} \neq 0, \tag{14}$$

*and so $(\mathbf{I} + \mathbf{H} \otimes \mathbf{A})$ does not act as a low-pass filter. If (3) multiple eigenvalues have the same dominating magnitude, and there is at least one dominating eigenvalue $(1 + \lambda_j^H \lambda_i^A)$ where $\lambda_i^A \neq \lambda_n^A$, then eq. (14) holds and $(\mathbf{I} + \mathbf{H} \otimes \mathbf{A})$ does not act as a low-pass filter.*

*Proof.* If (1), as $L \to \infty$ we have from Theorem 2 that,

$$\lim_{L \to \infty} \mathbf{X}_L = (1 + \lambda_j^H \lambda_n^A)^L s_{j,n} \mathbf{1} \mathbf{v}_j^{H^\top}. \tag{15}$$

If we plug this into the expression in Definition 1 of a low-pass filter we get,

$$
\begin{aligned}
\lim_{L \to \infty} \frac{\|\text{HFC}[\mathbf{X}_L]\|_2}{\|\text{LFC}[\mathbf{X}_L]\|_2} &= \lim_{L \to \infty} \sqrt{\frac{\|\text{HFC}[\mathbf{X}_L]\|_2^2}{\|\mathbf{X}_L - \text{HFC}[\mathbf{X}_L]\|_2^2}} \\
&= \lim_{L \to \infty} \sqrt{\frac{\|(\mathbf{I} - \frac{1}{n}\mathbf{1}\mathbf{1}^\top)\mathbf{X}_L\|_2^2}{\|\mathbf{X}_L - (\mathbf{I} - \frac{1}{n}\mathbf{1}\mathbf{1}^\top)\mathbf{X}_L\|_2^2}} \\
&= \lim_{L \to \infty} \sqrt{\frac{\|(\mathbf{I} - \frac{1}{n}\mathbf{1}\mathbf{1}^\top)(1 + \lambda_j^H \lambda_n^A)^L s_{j,n} \mathbf{1}\mathbf{v}_j^{H^\top}\|_2^2}{\|\mathbf{X}_L - (\mathbf{I} - \frac{1}{n}\mathbf{1}\mathbf{1}^\top)(1 + \lambda_j^H \lambda_n^A)^L s_{j,n} \mathbf{1}\mathbf{v}_j^{H^\top}\|_2^2}} \\
&= \lim_{L \to \infty} \sqrt{\frac{\|(1 + \lambda_j^H \lambda_n^A)^L s_{j,n}(\mathbf{1}\mathbf{v}_j^{H^\top} - \mathbf{1}\mathbf{v}_j^{H^\top})\|_2^2}{\|\mathbf{X}_L - (1 + \lambda_j^H \lambda_n^A)^L s_{j,n}(\mathbf{1}\mathbf{v}_j^{H^\top} - \mathbf{1}\mathbf{v}_j^{H^\top})\|_2^2}} \\
&= 0,
\end{aligned}
$$

where (17) is due to the fact that $(1/n)\mathbf{1}\mathbf{1}^\top \mathbf{M}$ averages the columns of any matrix $\mathbf{M} \in \mathbb{R}^{n\times r}$. This means that $(1/n)\mathbf{1}\mathbf{1}^\top \mathbf{1}\mathbf{v}_j^{H\top} = \mathbf{1}\mathbf{v}_j^{H\top}$ as $\mathbf{1}\mathbf{v}_j^{H\top}$ has identical values in each column.

If **(2)** we have from Theorem 2 that,

$$\lim_{L\to\infty} \mathbf{X}_L = (1+\lambda_j^H\lambda_1^A)^L s_{j,1}\mathbf{v}_1^A\mathbf{v}_j^{H\top}. \tag{16}$$

Plugging this into Definition 1 we get,

$$
\begin{aligned}
\lim_{L\to\infty}\frac{\|\mathrm{HFC}[\mathbf{X}_L]\|_2}{\|\mathrm{LFC}[\mathbf{X}_L]\|_2} &= \lim_{L\to\infty}\sqrt{\frac{\|\mathrm{HFC}[\mathbf{X}_L]\|_2^2}{\|\mathbf{X}_L - \mathrm{HFC}[\mathbf{X}_L]\|_2^2}}\\
&= \lim_{L\to\infty}\sqrt{\frac{\|(\mathbf{I}-\frac{1}{n}\mathbf{1}\mathbf{1}^\top)\mathbf{X}_L\|_2^2}{\|\mathbf{X}_L - (\mathbf{I}-\frac{1}{n}\mathbf{1}\mathbf{1}^\top)\mathbf{X}_L\|_2^2}}\\
&= \lim_{L\to\infty}\sqrt{\frac{\|(\mathbf{I}-\frac{1}{n}\mathbf{1}\mathbf{1}^\top)(1+\lambda_j^H\lambda_1^A)^L s_{j,1}\mathbf{v}_1^A\mathbf{v}_j^{H\top}\|_2^2}{\|\mathbf{X}_L - (\mathbf{I}-\frac{1}{n}\mathbf{1}\mathbf{1}^\top)(1+\lambda_j^H\lambda_1^A)^L s_{j,1}\mathbf{v}_1^A\mathbf{v}_j^{H\top}\|_2^2}}\\
&= \lim_{L\to\infty}\sqrt{\frac{\|(1+\lambda_j^H\lambda_1^A)^L s_{j,1}(\mathbf{v}_1^A\mathbf{v}_j^{H\top} - \frac{1}{n}\mathbf{1}\mathbf{1}^\top\mathbf{v}_1^A\mathbf{v}_j^{H\top})\|_2^2}{\|\mathbf{X}_L - (1+\lambda_j^H\lambda_1^A)^L s_{j,1}(\mathbf{v}_1^A\mathbf{v}_j^{H\top} - \frac{1}{n}\mathbf{1}\mathbf{1}^\top\mathbf{v}_1^A\mathbf{v}_j^{H\top})\|_2^2}}\\
&\neq 0.
\end{aligned}
$$

The final line holds because in general $(1/n)\mathbf{1}\mathbf{1}^\top\mathbf{v}_1^A\mathbf{v}_j^{H\top} \neq \mathbf{v}_1^A\mathbf{v}_j^{H\top}$, unless $\mathbf{v}_1^A = c\mathbf{1}$ for some $c \in \mathbb{R}$. However, this is impossible given Assumption 1, as the Perron-Frobenius Theorem states that there is only one eigenvector of $\mathbf{A}$ that has all positive real entries. As we know $\mathbf{v}_n^A = \mathbf{1}$, there is no other eigenvector of $\mathbf{A}$ such that $\mathbf{v}_i^A = c\mathbf{1}$. Therefore, $\lim_{L\to\infty}\frac{\|\mathrm{HFC}[\mathbf{X}_L]\|_2}{\|\mathrm{LFC}[\mathbf{X}_L]\|_2} > 0$.

If instead, **(3)**, then by the definition of $\mathcal{E}$ in Corollary 1 we have that,

$$\lim_{L\to\infty} \mathbf{X}_L = \sum_{(j,i)\in\mathcal{E}} (1+\lambda_j^H\lambda_i^A)^L s_{j,i}\mathbf{v}_i^A\mathbf{v}_j^{H\top}. \tag{17}$$

Therefore,

$$
\begin{aligned}
\lim_{L\to\infty}\frac{\|\mathrm{HFC}[\mathbf{X}_L]\|_2}{\|\mathrm{LFC}[\mathbf{X}_L]\|_2} &= \lim_{L\to\infty}\sqrt{\frac{\|\sum_{(j,i)\in\mathcal{E}}(1+\lambda_j^H\lambda_j^A)^L s_{j,1}(\mathbf{v}_i^A\mathbf{v}_j^{H\top} - \frac{1}{n}\mathbf{1}\mathbf{1}^\top\mathbf{v}_i^A\mathbf{v}_j^{H\top})\|_2^2}{\|\mathbf{X}_L - \sum_{(j,i)\in\mathcal{E}}(1+\lambda_j^H\lambda_j^A)^L s_{j,1}(\mathbf{v}_i^A\mathbf{v}_j^{H\top} - \frac{1}{n}\mathbf{1}\mathbf{1}^\top\mathbf{v}_i^A\mathbf{v}_j^{H\top})\|_2^2}}\\
&\neq 0.
\end{aligned}
$$

The final line follows so long as $\mathbf{v}_i^A\mathbf{v}_j^{H\top} \neq \mathbf{1}\mathbf{v}_j^{H\top}$ for at least one $(j,i) \in \mathcal{E}$. If this is true then we have one term in the sums above for which $(1/n)\mathbf{1}\mathbf{1}^\top\mathbf{v}_i^A\mathbf{v}_j^{H\top} \neq \mathbf{v}_i^A\mathbf{v}_j^{H\top}$. This is because $\mathbf{v}_i^A \neq c\mathbf{1}$ (by Assumption 1 and the Perron-Frobenius Theorem, as described in the proof of condition **(2)**). As we know that there is at least one dominating eigenvalue $(1+\lambda_j^H\lambda_i^A)$ where $\lambda_i^A \neq \lambda_n^A$ (this is given in the Theorem statement), then $\mathbf{v}_i^A\mathbf{v}_j^{H\top} \neq \mathbf{1}\mathbf{v}_j^{H\top}$, and so we are done. $\qquad\square$

## B  ADDITIONAL THEOREMS

**If $\lambda^H \in \mathbb{R}$.**  The following is a special case of Theorem 1 where all eigenvalues of $\mathbf{H}$ are real.

**Theorem 4** (eigenvalues $\lambda^H \in \mathbb{R}$). *Consider the Transformer update with fixed $\mathbf{A} > 0, \mathbf{H} := \mathbf{W}_{\mathrm{proj}}^\top\mathbf{W}_V^\top$, as described in eq. (3). Let $\{\lambda_i^A, \mathbf{v}_i^A\}_{i=1}^n$ and $\{\lambda_j^H, \mathbf{v}_j^H\}_{j=1}^d$ be the eigenvalue and eigenvectors of $\mathbf{A}$ and $\mathbf{H}$. Let the eigenvalues (and associated eigenvectors) be sorted in ascending order i.e., $\lambda_1^A \leq \cdots \leq \lambda_n^A$ and $\lambda_1^H \leq \cdots \leq \lambda_d^H$. Let the eigendecomposition of $(\mathbf{I} + \mathbf{W}_{\mathrm{proj}}^\top\mathbf{W}_V^\top \otimes \mathbf{A})$ be $\mathbf{Q}\Lambda\mathbf{Q}^{-1}$, where $\Lambda_{ji} = (1+\lambda_j^H\lambda_i^A)$. As the number of total layers $L \to \infty$, one of four possible*

*eigenvalues dominate the rest (multiple dominate if there are ties):*

$$
\begin{cases}
(1 + \lambda_d^H \lambda_n^A) & & \text{if } \lambda_j^H > 0, \forall j \in \{1, \dots, d\} \\
\begin{rcases}
(1 + \lambda_d^H \lambda_n^A) & & \text{if } \lambda_d^H + 2 > |\lambda_1^H| \\
(1 + \lambda_1^H \lambda_1^A) & \text{if } |1 + \lambda_1^H \lambda_1^A| > |1 + \lambda_1^H \lambda_n^A| \\
(1 + \lambda_1^H \lambda_n^A) & \text{if } |1 + \lambda_1^H \lambda_1^A| < |1 + \lambda_1^H \lambda_n^A|
\end{rcases} \; \text{if } \lambda_d^H + 2 < |\lambda_1^H| & \text{if } \lambda_1^H < 0, \lambda_d^H > 0 \\[6pt]
\begin{rcases}
(1 + \lambda_d^H \lambda_1^A) & \text{if } |1 + \lambda_d^H \lambda_1^A| > |1 + \lambda_1^H \lambda_n^A| \\
(1 + \lambda_1^H \lambda_n^A) & \text{if } |1 + \lambda_d^H \lambda_1^A| < |1 + \lambda_1^H \lambda_n^A|
\end{rcases} \text{if } \lambda_1^A > 0, \lambda_1^H > -2 \\
(1 + \lambda_1^H \lambda_n^A) & \text{if } \lambda_1^A > 0, \lambda_1^H < -2 \\
(1 + \lambda_1^H \lambda_1^A) & \text{if } \lambda_1^A < 0, \lambda_1^H > -2 \\
\begin{rcases}
(1 + \lambda_1^H \lambda_1^A) & \text{if } |1 + \lambda_1^H \lambda_1^A| > |1 + \lambda_1^H \lambda_n^A| \\
(1 + \lambda_1^H \lambda_n^A) & \text{if } |1 + \lambda_1^H \lambda_1^A| < |1 + \lambda_1^H \lambda_n^A|
\end{rcases} \text{if } \lambda_1^A < 0, \lambda_1^H < -2
\end{cases} \text{if } \lambda_j^H < 0, \forall j \in \{1, \dots, d\}
$$

*Proof.* Our goal is again to characterize the identity of $\lambda_{j^*}^H \lambda_{i^*}^A$ where $|1 + \lambda_{j^*}^H \lambda_{i^*}^A| > |1 + \lambda_{j'}^H \lambda_{i'}^A|$ for all $j' \in \{1, \dots, d\} \setminus j^*$ and $i' \in \{1, \dots, n\} \setminus i^*$, for all ranges of $\lambda_H, \lambda_A$. This is because $(1 + \lambda_{j^*}^H \lambda_{i^*}^A)^L$ will dominate as $L \to \infty$. We will again assume there are no ties i.e., $j^*, i^*$ each denote a single index. Given this, we detail each case described in the theorem statement.

**if $\lambda_j^H > 0$ for all $j \in \{1, \dots, d\}$.** First recall that $\lambda_i^A \in (-1, 1]$ and $\lambda_n^A = 1$. As $\lambda_j^H > 0$, we have that $|1 + \lambda_d^H \lambda_n^A| > |1 + \lambda_{j'}^H \lambda_{i'}^A|$ for all $j' \in \{1, \dots, d-1\}$ and $i' \in \{1, \dots, n-1\}$. This is because, by definition $\lambda_d^H \lambda_n^A > \lambda_{j'}^H \lambda_{i'}^A$. Further, $1 + \lambda_d^H \lambda_n^A > |1 + \lambda_{j'}^H \lambda_{i'}^A|$ as the largest $|1 + \lambda_{j'}^H \lambda_{i'}^A|$ can be is either (i) $|1 - \epsilon \lambda_d^H|$ for $0 < \epsilon < 1$ or (ii) $|1 + \lambda_{d-1}^H \lambda_n^A|$ (i.e., in (i) $\lambda_d^H$ is negated by $\lambda_1^A$ and in (ii) $\lambda_{d-1}^H$ is the next largest value of $\lambda^H$). For (i), it must be that $1 + \lambda_d^H \lambda_n^A > |1 - \epsilon \lambda_d^H|$ as $\lambda_d^H > 0$. For (ii) $\lambda_d^H > \lambda_{d-1}^H > 0$ (as we assume there are no ties), and so $|1 + \lambda_d^H \lambda_n^A| > |1 + \lambda_{d-1}^H \lambda_n^A|$. Therefore $(1 + \lambda_d^H \lambda_n^A)$ dominates.

**if $\lambda_1^H < 0, \lambda_d^H > 0$.** Recall we can view selecting $\lambda_j^H \lambda_i^A$ to maximize $|1 + \lambda_j^H \lambda_i^A|$ as maximizing distance to $-1$. In this condition the maximal $\lambda_{j^*}^H \lambda_{i^*}^A$ depends on whether $\lambda_1^H$ or $\lambda_d^H$ is farther away from $-1$. If $\lambda_d^H$ is farther from $-1$, i.e., $|1 + \lambda_d^H \lambda_n^A| > |1 + \lambda_1^H \lambda_n^A|$ (which can be simplified to $\lambda_d^H + 2 > |\lambda_1^H|$), then $|1 + \lambda_d^H \lambda_n^A|$ is maximal because (a) any other $\lambda_{i'}^A$ will move $\lambda_d^H$ closer to $-1$, and (b) any other $\lambda_{j'}^H$ is closer to $-1$. So $(1 + \lambda_d^H \lambda_n^A)$ dominates. If $\lambda_1^H$ is farther from $-1$, i.e., $|1 + \lambda_d^H \lambda_n^A| < |1 + \lambda_1^H \lambda_n^A|$ (which can be simplified to $\lambda_d^H + 2 < |\lambda_1^H|$), then it depends on whether $\lambda_1^A$ can push $\lambda_1^H$ farther away from $-1$ than $\lambda_1^H$ is itself (sidenote: this will only happen for $\lambda_1^A < 0$, when it can 'flip' $\lambda_1^H$ across the origin, because by definition it has to beat $\lambda_d^H > 0$). If it can, i.e., $|1 + \lambda_1^H \lambda_1^A| > |1 + \lambda_1^H \lambda_n^A|$ then $(1 + \lambda_1^H \lambda_1^A)$ dominates. Otherwise, $|1 + \lambda_1^H \lambda_1^A| < |1 + \lambda_1^H \lambda_n^A|$ and $(1 + \lambda_1^H \lambda_n^A)$ dominates.

**if $\lambda_j^H < 0$ for all $j \in \{1, \dots, d\}$.** In this case we need to know if (a) $\lambda_1^A > 0$ or (b) $\lambda_1^A < 0$. If (a) then all $\lambda_j^A > 0$ and so we cannot 'flip' $\lambda^H$ across the origin. Because of this, if $\lambda_1^H > -2$ then we have that $\lambda_j^H \lambda_i^A \in (-2, 0)$ for all $j$. Note that $|1 + \lambda_j^H \lambda_i^A|$ is symmetric in this interval around $-1$ so whichever $\lambda_j^H \lambda_i^A$ is closest to the ends of the interval will maximize $|1 + \lambda_j^H \lambda_i^A|$. Note that $\lambda_1^H \lambda_n^A$ will be closest to $-2$ and $\lambda_d^H \lambda_1^A$ will be closest to $0$. Therefore if $|1 + \lambda_d^H \lambda_1^A| > |1 + \lambda_1^H \lambda_n^A|$ then $(1 + \lambda_d^H \lambda_1^A)$ will dominate. If the opposite is true then $(1 + \lambda_1^H \lambda_n^A)$ will dominate. If instead $\lambda_1^H < -2$ then $\lambda_1^H \lambda_n^A$ is farthest from $-1$ as $\lambda_j^H \lambda_i^A < 0$, so $(1 + \lambda_1^H \lambda_n^A)$ dominates. If case (b) and we have that $\lambda_1^A < 0$ and $\lambda_1^H > -2$ then $\lambda_1^H \lambda_1^A > 0$. This means that $|1 + \lambda_1^H \lambda_1^A| > |1 + \lambda_j^H \lambda_i^A|$ because any 'flip' of $\lambda_j^H$ across the origin by $\lambda_i^A < 0$ makes $\lambda_j^H \lambda_i^A > \lambda_j^H \lambda_{i'}^A$ where $\lambda_{i'}^A > 0$. The flip that is largest is $\lambda_1^H \lambda_1^A > \lambda_j^H \lambda_i^A$, by definition of $\lambda_1^H, \lambda_1^A$. So $(1 + \lambda_1^H \lambda_1^A)$ dominates. If instead $\lambda_1^A < 0$ and $\lambda_1^H < -2$ Then it depends on whether $\lambda_1^A$ can 'flip' $\lambda_1^H$ farther from $-1$ than $\lambda_1^H$ is itself. If it can, then $(1 + \lambda_1^H \lambda_1^A)$ dominates, otherwise $(1 + \lambda_1^H \lambda_n^A)$ dominates. (For completeness, note that $\max\{|1 + \lambda_1^H \lambda_1^A|, |1 + \lambda_1^H \lambda_n^A|\} > |1 + \lambda_d^H \lambda_n^A|$ because either $\lambda_d^H \lambda_n^A < -2$ in which case

$|1 + \lambda_1^H \lambda_n^A| > |1 + \lambda_d^H \lambda_n^A|$ or $\lambda_d^H \lambda_n^A \in (-2, 0)$ in which case $|1 + \lambda_1^H \lambda_1^A| > |1 + \lambda_d^H \lambda_n^A|$. Also note that $|1 + \lambda_1^H \lambda_1^A| > |1 + \lambda_d^H \lambda_1^A|$ as $\lambda_d^H$ is closer to the origin than $\lambda_1^H$).

As these cases define a partition of $\lambda^H$ and $\lambda^A$, we are done. $\qquad\square$

## C ADDITIONAL RESULTS

Here we show the evolution of the average condition number of all $\mathbf{H}$ throughout training of ViT-Ti on CIFAR100.

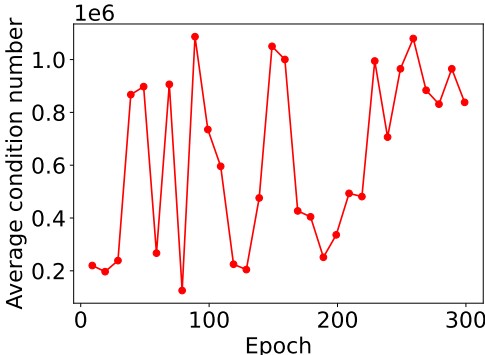

Figure 5: The average condition number of all $\mathbf{H}$ for ViT-Ti throughout training on CIFAR100.

