# OpenReview forum: "Setting the Record Straight on Transformer Oversmoothing"
_ICLR.cc/2024/Conference — Submitted to ICLR 2024_

### Official Review · Reviewer_yd6A · 2023-10-17

**Soundness:** 3 good
**Presentation:** 3 good
**Contribution:** 2 fair
**Rating:** 6
**Confidence:** 4

**Summary:**

The paper analyses the oversmoothing effect in Transformers. They find that Transformers are not inherently low-pass filters but oversmoothing depends on the eigenspectrum of the update equations. To prove this, the authors analyse a simplified Transformer architecture and incorporate the spectrum of attention and weight matrices. The authors relate their findings to existing pretrained architectures. They finally propose a reparametrization of the Transformer weights that ensures that oversmoothinng does not occur. The implications of such a reparametrization can be detrimental, as preliminary experiments show.

**Strengths:**

The paper builds on the established and successful framework of Wang et al, that analyses the degree to which a function is a low-pass filter. Doing so:
1. They give new insights on the eigenspectrum of both attention and weight matrices.
2. They find that, surprisingly, the transformer updates do not always lead to a low-pass filter.
3. Motivated by their findings, they propose a new parametrization for the linear layers following the self-attention computation. This leads to a change in the dominating eigenvalues. The authors showcase how these changes can lead to superior performance under cases of severe corruption or for low-data regimes.

**Weaknesses:**

1. Overall I find the main contributions/results hard to digest. Most of the results seem to be extensions from Wang et al.
2. Oversmoothing is primarily a problem that prohibits/hampers training at initialization (see e.g. Noci et al.). The authors make the same observation, while also noting that trained models exhibit different behavior. In that sense, it would make more sense to show if/how their new parametrization enables training in severe cases of oversmoothing, as e.g. when the networks become much deeper. Although I do find the applications of low-data and corruption interesting, their motivation is less clear. It could be interesting to analyse what is happening during the early phase of training.
3. Oversmoothing is less of a problem in ViT for image classification, compared to other scenarios. In image classification, labels are sparse -- 1 per sequence of tokens -- and some oversmoothing in deeper layers is expected -- in fact modern ViT just use the mean activations of the last layer to make predictions [1]. In that sense, a task in vision or language that requires a higher rank output, requiring a different prediction per token, could be more desirable.
4. The authors analyse a simplified Transformer architecture, making some non-trivial assumptions along the way. It is not clear how these findings generalize to a more general scenario. In more detail:
- They analyze a 1-head attention layer.
- They assume the same attention and weights are repeated across layers.
- They ignore the existence of LayerNorm.
- Attention weights change depending on the data.

Most notably, Pre-LN architectures [2] have been shown to effectively counteract some of the oversmoothing in Transformers. In the experiment sections, ViTs used by the authors seem to include LN as far as I can tell. The authors should make this clear.

[1] Zhai, Xiaohua, et al. "Scaling vision transformers." Proceedings of the IEEE/CVF Conference on Computer Vision and Pattern Recognition. 2022.

[2] Xiong, Ruibin, et al. "On layer normalization in the transformer architecture." International Conference on Machine Learning. PMLR, 2020.

**Questions:**

1. Can you comment on how your findings will change in the presence of different weights per layer and in the presence of pre-ln layers?
2. If oversmoothing is the problem, what about other ways to mitigate it? I am talking about scaling the residuals (Noci et al, [1, 2]) or initializing the attention layers differently, e.g. [3] or different initializations per layer, e.g. [4]. There is a long list of proposed techniques in the literature from the signal propagation perspective. Since you are proposing a new parametrization, it makes sense to compare what you are achieving compares to what they are trying to achieve. Ensuring that the Transformer is not a low-pass filter, does not necessarily mean that any of the meaningful signal is preserved or that feature learning can take place.
3. Before section 5, should the superscript $^+$ model be initialized as $\text{diag}(\Lambda_H) = + (\psi^2)$?
4. Can you comment on the stability of your new parametrization? Especially what (if any) are the differences in the early stage of training.

[1] Noci, Lorenzo, et al. "The shaped transformer: Attention models in the infinite depth-and-width limit." arXiv preprint arXiv:2306.17759 (2023).

[2] He, Bobby, et al. "Deep transformers without shortcuts: Modifying self-attention for faithful signal propagation." arXiv preprint arXiv:2302.10322 (2023).

[3] Trockman, Asher, and J. Zico Kolter. "Mimetic Initialization of Self-Attention Layers." arXiv preprint arXiv:2305.09828 (2023).

[4] Zhang, Hongyi, Yann N. Dauphin, and Tengyu Ma. "Fixup initialization: Residual learning without normalization." arXiv preprint arXiv:1901.09321 (2019).

---

> ### Author Response · Authors · 2023-11-19
> **Author Response to Reviewer yd6A (Part 1)**
>
> Dear Reviewer yd6A,
>
> Thank you so much for your time and many comments. We respond to each question below.
>
> > [..Weaknesses:..]
>
> > [..1. Overall I find the main contributions/results hard to digest. Most of the results seem to be extensions from Wang et al...]
>
> Thank you for asking about this. We completely agree that Wang et al., 2022 provides big inspiration for this work. Their analysis is extremely insightful and our goal is to further extend this analysis to uncover the relationship between the eigenspectrum of the weights and the suppression (or not) of high frequencies when the weights H and residual connection are taken into account. This turns out to be very important as it gives us tools to control the eigenspectrum of the weights in order to induce smoothing or sharpening.
>
> Beyond the nice work of Wang et al., 2022, we were also heavily inspired by the work of [Dong et al., Attention is not all you need: Pure attention loses rank doubly exponentially with depth (ICML 2021)] who show that feature representations in Transformers can rank collapse to rank 1, but that there exist infinitely many weights that do not rank collapse with the residual connection. We extend this to show exactly which weights do not rank collapse: only when multiple eigenvalues $(1 + \lambda^H_j \lambda^A_i)$ simultaneously have equivalent dominant magnitudes. However, this is a rare case: because $\mathbf{A}$ and $\mathbf{H}$ are learned, it is unlikely the magnitude $|1 + \lambda^H_j \lambda^A_i|$ of any two eigenvalues will be identical.
>
> We were also influenced by the extensive analysis of [Park & Kim How do vision transformers work? (ICLR 2022)]: we base our experiment setup off of theirs. Finally, our inspiration to look deeper into how the eigenspectrum can be connected to smoothing, while including the weights and residual connection is also inspired by the related line of work on oversmoothing in graph neural networks e.g., [Di Giovanni et al., Understanding convolution on graphs via energies. (TMLR 2023)].
>
> > [..2. show if/how their new parametrization enables training in severe cases of oversmoothing, as e.g. when the networks become much deeper...]
>
> Thank you for raising this point, we agree. To answer this we have added an experiment where we double the depth of the ViT-Ti models on CIFAR100. We have also added a parameterization variant where the first $L/2$ layers sharpen (as in ViT-Ti$^-$) and the remaining $L/2$ layers are left normal (as in ViT-Ti), which we call ViT-Ti*. Finally we have added a comparison FeatScale by Wang et al., 2022. All results are averaged over 3 trials with standard deviations (we bold the best result in each column and any method whose mean plus standard deviation is greater than or equal to mean of the best result):
>
> [new] Increased Depth (Table 3)
>
> | Depth       	| 12       	| 24       	|
> |----------------|--------------|--------------|
> | ViT-Ti     	| 66.78 ± 0.1  | 67.22 ± 0.1  |
> | ViT-Ti + FeatScale | 66.57 ± 0.5  | 67.71 ± 0.5  |
> | ViT-Ti$^+$ 	| 64.34 ± 0.3  | 65.65 ± 0.3 |
> | ViT-Ti$^-$ 	| 66.62 ± 0.3  | 66.76 ± 0.3  |
> | ViT-Ti$^*$ 	| **67.84 ± 0.6**  | **69.05 ± 0.1**  |
>
> Here ViT-Ti$^+$ and ViT-Ti$^-$ both underperform ViT-Ti, and we suspect this is because ViT-T$^-$ sharpens too much, amplifying noise in the data (whereas ViT-Ti$^+$ oversmooths the data). On the other hand ViT-Ti* significantly outperforms all other models not just for larger depths in Table 3, but also in corruption robustness (Table 4), and data efficiency (Table 2). This suggests that both oversmoothing and oversharpening are detrimental to performance, and that a careful balance is key to an accurate model. We have updated the paper to describe these results and look forward to investigating this further in future work.
>
> > [..3. ...a task in vision or language that requires a higher rank output, requiring a different prediction per token, could be more desirable...]
>
> This is a very good point. We are interested to try out the approach in a more label-dense vision task such as segmentation, where oversmoothing poses much larger problems than image classification as you suggest.
>
> (response continues below)

---

> ### Author Response · Authors · 2023-11-19
> **Author Response to Reviewer yd6A (Part 2)**
>
> > [..4. ...The authors analyse a simplified Transformer architecture...Pre-LN architectures [2] have been shown to effectively counteract some of the oversmoothing...]
>
> > [..Questions:..]
>
> > [..1. ...how your findings will change in the presence of different weights per layer and in the presence of pre-ln layers?...]
>
> We were also interested in understanding how our theoretical results generalize to a more general scenario with changing parameters and Pre-LN architectures. To test this, in our experiments we allow both attention matrices $\mathbf{A}$ and weights $\mathbf{H}$ to change every layer, and we use the same architecture as ViT-Ti (a Pre-LN architecture). We were excited to see that even though these models don’t follow the assumptions of our theoretical results, we can still use them to control the smoothing/sharpening behavior of these models. Thank you for brining this up. We have clarified these details in the paper.
>
> > [..2. If oversmoothing is the problem, what about other ways to mitigate it? I am talking about scaling the residuals (Noci et al, [1, 2]) or initializing the attention layers differently, e.g. [3] or different initializations per layer, e.g. [4]...]
>
> Thanks for bringing this up. One of the strengths of our approach is that it is complementary to these approaches (scaling residuals [1 ,2], different initializations [3, 4]) and so we can apply them at the same time as our approach. Because of this we opted to compare against an approach that isn’t complementary, specifically FeatScale by Wang et al., 2022. Alongside the above increased depth experiment we also compare its data efficiency and corruption robustness below:
>
> Data Efficiency (Table 2)
>
> | Data Kept          	| 100%     	| 50%      	| 10%      	|
> |------------------|--------------|--------------|--------------|
> | ViT-Ti       	| 66.78 ± 0.1  | 56.38 ± 0.5  | 32.47 ± 0.9  |
> | ViT-Ti + FeatScale | 66.57 ± 0.5  | 57.08 ± 0.3  | 32.50 ± 0.4  |
> | ViT-Ti$^+$   	| 64.34 ± 0.3  | 55.21 ± 0.5  | 28.67 ± 0.5  |
> | ViT-Ti$^-$   	| 66.62 ± 0.3  | 56.58 ± 0.3  | **33.99 ± 0.4**  |
> | ViT-Ti$^*$   	| **67.84 ± 0.6**  | **57.78 ± 0.4**  | **33.64 ± 0.5**  |
>
>
> Corruption Robustness (Table 4)
>
> | Corruption Intensity | 0          	| 1          	| 2          	| 3          	| 4          	| 5          	|
> |----------------------|----------------|----------------|----------------|----------------|----------------|----------------|
> | ViT-Ti           	| 66.78 ± 0.1	| 61.59 ± 0.2	| 57.75 ± 0.2	| 53.12 ± 0.1	| 49.32 ± 0.1	| 39.17 ± 0.2	|
> | ViT-Ti + Featscale   | 66.57 ± 0.5	| 61.25 ± 0.5	| 57.36 ± 0.6	| 52.64 ± 0.6	| 48.87 ± 0.5	| 38.77 ± 0.4	|
> | ViT-Ti$^+$       	| 64.34 ± 0.3	| 58.36 ± 0.4	| 53.83 ± 0.5	| 49.03 ± 0.6	| 45.15 ± 0.6	| 35.86 ± 0.6	|
> | ViT-Ti$^-$       	| 66.62 ± 0.3	| 61.62 ± 0.2	| 58.00 ± 0.3	| 53.56 ± 0.2	| 49.70 ± 0.3	| 40.07 ± 0.2	|
> | ViT-Ti$^*$       	| **67.84 ± 0.6**	| **62.49 ± 0.5**	| **58.82 ± 0.7**	| **54.32 ± 0.7**	| **50.40 ± 0.6**	| **40.38 ± 0.7**	|
>
> > [..3. Before section 5, should the superscript $\mbox{}^+$ model be initialized as $\mbox{diag}(\Lambda_H) = +(\psi^2)$?..]
>
> This is correct! Thank you for catching this typo!
>
> > [..4 . Can you comment on the stability of your new parametrization? Especially what (if any) are the differences in the early stage of training..]
>
> This is also interesting! To better understand what is happening early in training we have added a plot of HFC/LFC for the final layer against training iterations in Figure 4 in Section 5. Interestingly, we see that the last layers of both ViT-Ti$^*$ and ViT-Ti have the same filtering properties from the beginning to the end of training, despite the fact that the first $L/2$ layers of ViT-Ti$^*$ are sharpening. Further at the beginning of training, the last layer of ViT-Ti$^-$ becomes less sharp, but by the end of training it becomes sharper than at initialization. Thank you for asking about this and for these citations, we will add them to the paper.
>
> Thank you again for your time. Let us know if you have any other questions and we will respond as soon as possible.

---

> > ### Comment · Reviewer_yd6A · 2023-11-20
> >
> > Thank you very much for the clarifications and the additional results. I am replying using the same order as you did.
> >
> > > Contributions of the paper
> >
> > I appreciate your honesty and the more precise list of inspiration for your work. I summarize your contribution as "We extend this to show exactly which weights do not rank collapse". This is indeed interesting, although Theorem 1 as I said is not easily parsable. I am not arguing that every theorem should be, but you can perhaps think of a different way to formulate it.
> >
> > > Other ways to mitigate oversmoothing.
> >
> > Although I do find some merit in the "ViT-Ti*" model, it seems more like you are overfitting this single classification dataset. I do not believe that you should necessarily get the best results, but instead focus on cases where training is significantly obscured, as when going even deeper. If not such cases exist, this damages the motivation of your study.
> >
> > > label, dense task
> >
> > > generalization to the more comple Transformer architecture
> >
> > Still seems like an important question to me
> >
> > > Other baselines
> >
> > I do not fully agree that all these methods are complementary. Sure, you can apply them in parallel. But if you just want to avoid oversmoothing, one of them is enough.
> >
> > > Stability of new parametrization
> >
> > Appreciate the new results. It seems that there are many unanswered questions.
> >
> > All in all, I appreciate your commitment. I believe there are many unanswered questions. The main takeaway message is nonetheless clear message: not all weights cause rank collapse.

---

> > > ### Author Response · Authors · 2023-11-21
> > > **Author Response to Reviewer yd6A (Part 3)**
> > >
> > > Thank you again for your additional comments. We respond to each of them below.
> > >
> > > > [..I summarize your contribution as "We extend this to show exactly which weights do not rank collapse"..]
> > >
> > > Another contribution is our eigenspectrum analysis in Theorems 1, 2, and 3 and Corollary 2. This reveals that we can parameterize weights $\mathbf{H}$ to directly control smoothing or sharpening. We show that this parameterization improves generalization and continues to do so when training data is decreased (Table 2), depth is increased (Table 3), and data corruption is increased (Table 4). As far as we are aware we are the first to investigate the relationship between sharpening and data efficiency and corruption robustness.
> > >
> > > > [..This is indeed interesting, although Theorem 1 as I said is not easily parsable. I am not arguing that every theorem should be, but you can perhaps think of a different way to formulate it..]
> > >
> > > Thanks for this. Based on this suggestion we have simplified Theorem 1 and updated the paper. We have also added Corollary 3 and Theorem 4 for more specific cases of $\mathbf{H}$ to provide additional intuition.
> > >
> > > > [..Although I do find some merit in the "ViT-Ti*" model, it seems more like you are overfitting this single classification dataset.  I do not believe that you should necessarily get the best results,..]
> > >
> > > > [..label, dense task..]
> > >
> > > Thanks. We will also test on ImageNet and include the results in the final version. Once an ImageNet backbone is available we will also be able to use it for a broader range of tasks. Specifically, we will test it on the label-dense task of image segmentation.
> > >
> > > > [..focus on cases where training is significantly obscured, as when going even deeper..]
> > >
> > > We have run a new experiment where we increase the network depth. We have listed this in the Part 1 response to your review, and Table 3 in the updated paper.
> > >
> > > > [..generalization to the more complete Transformer architecture..]
> > >
> > > We believe there is a slight confusion. In all of the experiments we used a complete Transformer architecture where both attention matrices $\mathbf{A}$ and weights $\mathbf{H}$ change every layer. We use the same architecture as ViT-Ti, a Pre-LN architecture.
> > >
> > > > [..I do not fully agree that all these methods are complementary. Sure, you can apply them in parallel. But if you just want to avoid oversmoothing, one of them is enough..]
> > >
> > > We disagree here. We have evidence that using one method to avoid oversmoothing is not enough. Specifically, we use Pre-LN architectures in all our experiments, which as you point out, has been shown to counteract oversmoothing in Transformers. However, as shown in Figure 3, we can make these models oversmooth by parameterizing $\mathbf{H}$ with positive eigenvalues. Additionally, all of the Pre-LN models trained on ImageNet start to oversmooth halfway through the network in Figure 3 (c). So one method to prevent oversmoothing is not always enough.
> > >
> > > > [..Stability of new parametrization..]
> > >
> > > > [..Appreciate the new results. It seems that there are many unanswered questions..]
> > >
> > > > [..All in all, I appreciate your commitment. I believe there are many unanswered questions..]
> > >
> > > Thank you. If you have any more unanswered questions we will respond as soon as possible.

---

> > > > ### Comment · Reviewer_yd6A · 2023-11-21
> > > >
> > > > Thank you for the additional details. I think my concerned are mostly covered with either additional explanations or promises for future experiments.
> > > >
> > > > > [..focus on cases where training is significantly obscured...]
> > > >
> > > > As you point out, for up to 24 layers, ViT-Ti$^{-}$ and ViT-Ti have negligible difference in performance in Table 3, so I don't see how this proves your point. Oversmoothing might be bad at the beginning of training, as proven by Dong et al, Noci et al, but further during training is hard to say anything. I would thus focus more on initialization.
> > > >
> > > > > [..generalization to the more complete Transformer architecture..]
> > > >
> > > > My point was about the theory and not the empirical results.
> > > >
> > > > > [...methods are complementary...]
> > > >
> > > > Although pre-LN helps with oversmoothing, I never suggested this as a baseline. Especially I proposed "I am talking about scaling the residuals (Noci et al, [1, 2]) or initializing the attention layers differently, e.g. [3] or different initializations per layer, e.g. [4]".

---

> > > > > ### Author Response · Authors · 2023-11-21
> > > > > **Author Response to Reviewer yd6A (Part 4)**
> > > > >
> > > > > Thank you for these clarifications! We respond to each below.
> > > > >
> > > > > > [..As you point out, for up to 24 layers, ViT-Ti and ViT-Ti have negligible difference in performance in Table 3..]
> > > > >
> > > > > Agreed! However, ViT-Ti* significantly outperforms all other methods (omitting the '-NS' variants).
> > > > >
> > > > > > [..My point was about the theory and not the empirical results..]
> > > > >
> > > > > Ah understood! We agree this is a very interesting line of future work. This has so far proven elusive as even introducing say multiple heads requires us to relate the eigenvalues of each head to the eigenvalues of the sum of all heads. However, we believe that we may be able to find a solution by leveraging the results of [Klyachko, Alexander. "Vector bundles, linear representations, and spectral problems." (2003)].
> > > > >
> > > > > > [..Oversmoothing might be bad at the beginning of training, as proven by Dong et al, Noci et al, but further during training is hard to say anything. I would thus focus more on initialization..]
> > > > >
> > > > > > [..Although pre-LN helps with oversmoothing, I never suggested this as a baseline. Especially I proposed "I am talking about scaling the residuals (Noci et al, [1, 2]) or initializing the attention layers differently, e.g. [3] or different initializations per layer, e.g. [4]"..]
> > > > >
> > > > > Thank you for this. We agree this is interesting. Based on your comments we will also compare against at least one of the above methods.
> > > > >
> > > > > Thank you again for your feedback.

---

> ### Author Response · Authors · 2023-11-22
> **Author Comment to Reviewer yd6A**
>
> Dear Reviewer yd6A,
>
> Thank you again for your time on the paper. Do you have any extra questions about the paper or our response?
>
> Let us know and we will try to response as soon as possible before the discussion period ends on November 22.
>
> Thank you,
>
> The authors

---

### Official Review · Reviewer_rqLo · 2023-10-31

**Soundness:** 3 good
**Presentation:** 3 good
**Contribution:** 3 good
**Rating:** 6
**Confidence:** 4

**Summary:**

This paper studies the oversmoothing behavior in deep transformer networks. The authors' main focus is in showing under what conditions popular transformer architectures are amenable to oversmoothing and how to mitigate it. The authors mainly analyze residual plus self attention layer using its eigenspectrum. They first show that one eigenvalue will dominate the rest based on phases of H and eigenvalues of A. Depending on the dominating eigenvalue, the representation of the network will converge either to a single vector or a rank one matrix. Theorem-3 further shows that oversmoothing is not inevitable and residual connections and H can counteract it. Finally, the authors show a sufficient condition for counteracting oversmoothing via constraining the eigenspectrum of H to [-1, 0). Experimental results show that proposed solution is effective in counteracting oversmoothing and existing networks do not necessarily oversmooth.

**Strengths:**

The paper is easy to follow with necessary details given to understand the theory. It extends the available theory and proposes a new perspective on reparameterization. Experimental results support the proposed theory and preventive measures on oversmoothing.

**Weaknesses:**

While the theory extends previous work in some aspects, it is not clear if this difference is significant. Other preventive measures such as AttnScale/FeatScale from Wang et al. needs comparison and more discussion. Several details are also missing from the paper.

1. Wang et al also examines residual blocks and FFN in addition to the self-attention layer. While they mention the inevitability of high frequency components to be suppressed, their analysis suggest that using residual connections might prevent it from collapsing to zero. In fact, Eq (6) in Theorem-3 suggests that it is possible to prevent decaying of high-frequency components by constraining the $||W_V||_2$ properly so that it is a non-contractive mapping. I think a more detailed comparison to Wang et al is needed to highlight how your analysis differs from low-pass filtering aspect and how your preventive measures are different from constraining $||W_V||_2$ or AttnScale/FeatScale that Wang et al applies.

2. While Figure-2 shows that asymmetry degrades with more epochs, is 0.7 small enough to suggest symmetry, given that it starts at 0.95? What about outliers in the off-diagonal entries? Additionally, can you connect it to the condition number?

3. Can you clarify more on how do you update $\Theta$ in $QR(\Theta)=[V_H, R]$? Do you do QR decomposition after every gradient step or do you use $\Theta R^{-1}=V_H$ where you backpropagate gradients directly to $\Theta$?

4. How does the HFC/LFC evolve in training steps?

5. At the end of Section 4, page 6, why do you have $V_H^{-1}$ after clip?

6. The same page, last paragraph, I think it should be $diag(\Lambda_H)=\Psi^2$ -- no negative sign.

7. Please define Q in the main statement of Theorem-2.

8. Page 6, "||" is missing in definition of asymmetry.

9. Please describe the metrics for Table-2 and Table-3.

**Questions:**

Please see above for more details.

1. Can you provide a more detailed comparison with Wang et al? Including more baselines.

2. How should I interpret the asymmetry metric? How does it relate to condition number?

3. Can you give more details on updating QR decomposition?

4. How does HFC/LFC evolve during training?

---

> ### Author Response · Authors · 2023-11-18
> **Author Response to Reviewer rqLo (Part 1)**
>
> Dear Reviewer rqLo,
>
> Thank you so much for your time and detailed response. We respond to each question below.
>
> > [..Weaknesses:..]
>
> > [..1. ...a more detailed comparison to Wang et al is needed..]
>
> Thanks for this! We agree that it would be enlightening to compare with Wang et al., 2022 and so we have implemented their FeatScale method. Here are the new accuracy results. We have added another a new experiment where we double the model depth (Table 3) and a parameterization variant where the first $L/2$ layers sharpen (as in ViT-Ti$^-$) and the remaining $L/2$ layers are left normal (as in ViT-Ti), which we call ViT-Ti*. All results are averaged over 3 trials with standard deviations (we bold the best result in each column and any method whose mean plus standard deviation is greater than or equal to mean of the best result):
>
> Data Efficiency (Table 2)
>
> | Data Kept          	| 100%     	| 50%      	| 10%      	|
> |------------------|--------------|--------------|--------------|
> | ViT-Ti       	| 66.78 ± 0.1  | 56.38 ± 0.5  | 32.47 ± 0.9  |
> | ViT-Ti + FeatScale | 66.57 ± 0.5  | 57.08 ± 0.3  | 32.50 ± 0.4  |
> | ViT-Ti$^+$   	| 64.34 ± 0.3  | 55.21 ± 0.5  | 28.67 ± 0.5  |
> | ViT-Ti$^-$   	| 66.62 ± 0.3  | 56.58 ± 0.3  | **33.99 ± 0.4**  |
> | ViT-Ti$^*$   	| **67.84 ± 0.6**  | **57.78 ± 0.4**  | **33.64 ± 0.5**  |
>
>
> [new] Increased Depth (Table 3)
>
> | Depth       	| 12       	| 24       	|
> |----------------|--------------|--------------|
> | ViT-Ti     	| 66.78 ± 0.1  | 67.22 ± 0.1  |
> | ViT-Ti + FeatScale | 66.57 ± 0.5  | 67.71 ± 0.5  |
> | ViT-Ti$^+$ 	| 64.34 ± 0.3  | 65.65 ± 0.3 |
> | ViT-Ti$^-$ 	| 66.62 ± 0.3  | 66.76 ± 0.3  |
> | ViT-Ti$^*$ 	| **67.84 ± 0.6**  | **69.05 ± 0.1**  |
>
>
> Corruption Robustness (Table 4)
>
> | Corruption Intensity | 0          	| 1          	| 2          	| 3          	| 4          	| 5          	|
> |----------------------|----------------|----------------|----------------|----------------|----------------|----------------|
> | ViT-Ti           	| 66.78 ± 0.1	| 61.59 ± 0.2	| 57.75 ± 0.2	| 53.12 ± 0.1	| 49.32 ± 0.1	| 39.17 ± 0.2	|
> | ViT-Ti + Featscale   | 66.57 ± 0.5	| 61.25 ± 0.5	| 57.36 ± 0.6	| 52.64 ± 0.6	| 48.87 ± 0.5	| 38.77 ± 0.4	|
> | ViT-Ti$^+$       	| 64.34 ± 0.3	| 58.36 ± 0.4	| 53.83 ± 0.5	| 49.03 ± 0.6	| 45.15 ± 0.6	| 35.86 ± 0.6	|
> | ViT-Ti$^-$       	| 66.62 ± 0.3	| 61.62 ± 0.2	| 58.00 ± 0.3	| 53.56 ± 0.2	| 49.70 ± 0.3	| 40.07 ± 0.2	|
> | ViT-Ti$^*$       	| **67.84 ± 0.6**	| **62.49 ± 0.5**	| **58.82 ± 0.7**	| **54.32 ± 0.7**	| **50.40 ± 0.6**	| **40.38 ± 0.7**	|
>
> We have added these results to the paper. We suspect the reason that FeatScale underperforms is because it is still motivated assuming high frequencies are always suppressed, specifically they (Wang et al., 2022) say “According to our analysis in Section 2.2, MSA module will indiscriminately suppress high-frequency signals, which leads to severe information loss. Even though residual connection can retrieve lost information through the skip path, the high-frequency portion will be inevitably diluted (Theorem1). To this end, we propose another scaling technique that operates on feature maps, named Feature Scaling (FeatScale).”
> We agree with you that their analysis suggests that residual connections might prevent high frequencies from collapsing to zero, but Wang et al., 2022 disagree: “Although multi-head, FFN, and skip connection all help preserve the high-frequency signals, none would change the fact that MSA block as a whole only possesses the representational power of low-pass filters”.
> We believe the reason they argue this is because when they analyze the update equations that include residual connections (Proposition 5) they do not account for the eigenspectrum of those equations. Our goal with this work is to set the record straight with a new analysis that uncovers the relationship between the eigenspectrum of the weights and the suppression (or not) of high frequencies in the update equations.
>
> > [..2. is 0.7 small enough to suggest symmetry, given that it starts at 0.95?..]
>
> Thank you for bringing this up. It’s true that Figure 2 does not suggest perfect symmetry. Our original motivation for using a symmetric parameterization was inspired by [Hu et al., 2019. Exploring weight symmetry in deep neural networks.] who show that weight symmetry in NNs does not affect their universal approximation capabilities. However, we agree that it would be interesting to test a non-symmetric parameterization. Based on your suggestion we have implemented a new parameterization for $\mathbf{H}$ as $\mathbf{H} = \mathbf{V}_H \Lambda \mathbf{V}_H^{-1}$. Specifically, we take gradients to $\mathbf{V}_H$ in the backwards pass, and compute $\mathbf{H}$ using the above equation in the forwards pass. We use the suffix “-NS” for the non-symmetric parameterizations. Here are the new accuracy results:
>
> (response continues below)

---

> ### Author Response · Authors · 2023-11-18
> **Author Response to Reviewer rqLo (Part 2)**
>
> Data Efficiency (Table 2)
>
> | Data Kept          	| 100%     	| 50%      	| 10%      	|
> |------------------|--------------|--------------|--------------|
> | ViT-Ti       	| 66.78 ± 0.1  | 56.38 ± 0.5  | 32.47 ± 0.9  |
> | ViT-Ti + FeatScale | 66.57 ± 0.5  | 57.08 ± 0.3  | 32.50 ± 0.4  |
> | ViT-Ti$^+$   	| 64.34 ± 0.3  | 55.21 ± 0.5  | 28.67 ± 0.5  |
> | ViT-Ti$^-$   	| 66.62 ± 0.3  | 56.58 ± 0.3  | **33.99 ± 0.4**  |
> | ViT-Ti$^*$   	| 67.84 ± 0.6  | **57.78 ± 0.4**  | **33.64 ± 0.5**  |
> | ViT-Ti-NS$^-$	| 67.80 ± 0.2  | 57.54 ± 0.1  | **33.78 ± 0.3**  |
> | ViT-Ti-NS$^*$	| **68.53 ± 0.6**  | **57.67 ± 0.3**  | 33.75 ± 0.2  |
>
>
> [new] Increased Depth (Table 3)
>
> | Depth       	| 12       	| 24       	|
> |----------------|--------------|--------------|
> | ViT-Ti     	| 66.78 ± 0.1  | 67.22 ± 0.1  |
> | ViT-Ti + FeatScale | 66.57 ± 0.5  | 67.71 ± 0.5  |
> | ViT-Ti$^+$ 	| 64.34 ± 0.3  | 65.65 ± 0.3 |
> | ViT-Ti$^-$ 	| 66.62 ± 0.3  | 66.76 ± 0.3  |
> | ViT-Ti$^*$ 	| 67.84 ± 0.6  | 69.05 ± 0.1  |
> | ViT-Ti-NS$^-$  | 67.80 ± 0.2  | 67.09 ± 0.1  |
> | ViT-Ti-NS$^*$  | **68.53 ± 0.6**  | **69.64 ± 0.3**  |
>
>
> Corruption Robustness (Table 4)
>
> | Corruption Intensity | 0          	| 1          	| 2          	| 3          	| 4          	| 5          	|
> |----------------------|----------------|----------------|----------------|----------------|----------------|----------------|
> | ViT-Ti           	| 66.78 ± 0.1	| 61.59 ± 0.2	| 57.75 ± 0.2	| 53.12 ± 0.1	| 49.32 ± 0.1	| 39.17 ± 0.2	|
> | ViT-Ti + Featscale   | 66.57 ± 0.5	| 61.25 ± 0.5	| 57.36 ± 0.6	| 52.64 ± 0.6	| 48.87 ± 0.5	| 38.77 ± 0.4	|
> | ViT-Ti$^+$       	| 64.34 ± 0.3	| 58.36 ± 0.4	| 53.83 ± 0.5	| 49.03 ± 0.6	| 45.15 ± 0.6	| 35.86 ± 0.6	|
> | ViT-Ti$^-$       	| 66.62 ± 0.3	| 61.62 ± 0.2	| 58.00 ± 0.3	| 53.56 ± 0.2	| 49.70 ± 0.3	| 40.07 ± 0.2	|
> | ViT-Ti$^*$       	| 67.84 ± 0.6	| 62.49 ± 0.5	| 58.82 ± 0.7	| 54.32 ± 0.7	| 50.40 ± 0.6	| 40.38 ± 0.7	|
> | ViT-Ti-NS$^-$    	| 67.80 ± 0.2	| **63.03 ± 0.2**	| **59.65 ± 0.1**	| **55.05 ± 0.3**	| **51.30 ± 0.1**	| **41.33 ± 0.1**	|
> | ViT-Ti-NS$^*$    	| **68.53 ± 0.6**	| 63.01 ± 0.1	| **59.44 ± 0.3**	| **54.90 ± 0.4**	| **51.12 ± 0.4**	| 40.90 ± 0.4	|
>
> We have added these results to the paper. The non-symmetric parameterization does improve upon the symmetric parameterization in some cases (particularly for increased depth and corrupted data). We appreciate your suggestion to investigate this.
>
> > [..Additionally, can you connect it to the condition number?...How should I interpret the asymmetry metric? How does it relate to condition number?..]
>
> This is an interesting question. To investigate this we computed the average condition number for all H matrices used in ViT-Ti for every epoch in training on CIFAR100. We have added the results to Appendix C. The condition number fluctuates during training but gradually increases throughout training. We do not know of a connection between the condition number and the asymmetry of H, but agree it would be an interesting direction for future work.
>
> > [..3. how do you update $\Theta$ in $QR(\Theta) = [V_H, R]$?...Can you give more details on updating QR decomposition?..]
>
> We backpropagate through the QR decomposition, which requires a QR decomposition for each gradient step. This is because we require $V_H$ to be orthogonal. If we instead parameterized $V_H = \Theta R^{-1}$ and used gradient descent to update $\Theta$ then this could break the orthogonality of $V_H$. Thank you for this, we will add this detail to make this clearer in the paper.
>
> > [..4. How does the HFC/LFC evolve in training steps?..]
>
> Good question. To answer this we plot the HFC/LFC of the final layer for every epoch during training for the best performing symmetric models (ViT-Ti$^-$ and ViT-Ti$^*$), as well as ViT-Ti, and present the results in Figure 4 in Section 5. Interestingly, we see that the last layers of both ViT-Ti$^*$ and ViT-Ti have the same filtering properties throughout training, despite the fact that the first $L/2$ layers of ViT-Ti$^*$ are sharpening. Further at the beginning of training, the last layer of ViT-Ti$^-$ becomes less sharp, but by the end of training it becomes sharper than at initialization. Thank you for this suggestion!
>
> > [..5. why do you have $V_H^{-1}$ after clip?..]
>
> > [..6. I think it should be $\mbox{diag}(\Lambda_H) = \Phi^2$..]
>
> > [..8. "||" is missing in definition of asymmetry..]
>
> Thank you for catching these! There should be no $V_H^{-1}$, no negative sign, and additional ||.
>
> > [..7. Please define $Q$ in the main statement of Theorem-2..]
>
> Good call, we have moved the definition from the proof to the theorem statement.
>
> > [..9. Please describe the metrics for Table-2 and Table-3..]
>
> Sorry, these are test accuracies, thank you for asking!
>
> Thank you again for your time. Let us know if you have any other questions and we will respond as soon as possible.

---

> ### Author Response · Authors · 2023-11-22
> **Author Comment to Reviewer rqLo**
>
> Dear Reviewer rqLo,
>
> Thank you again for your time on the paper. Do you have any extra questions about the paper or our response?
>
> Let us know and we will try to response as soon as possible before the discussion period ends on November 22.
>
> Thank you,
>
> The authors

---

### Official Review · Reviewer_m7iA · 2023-11-01

**Soundness:** 3 good
**Presentation:** 3 good
**Contribution:** 3 good
**Rating:** 6
**Confidence:** 4

**Summary:**

This paper delves into the issue of oversmoothing in Transformers, demonstrating that it is not a pervasive problem, as Transformers do not always act as low-pass filters. The authors provide a theoretical analysis to delineate the conditions under which oversmoothing occurs and when it does not. Drawing insights from these findings, they introduce a novel reparameterization technique designed to mitigate oversmoothing. In addition, the authors conducted experiments with the Vision Transformer (VIT) and showcased the effectiveness of their proposed approach in addressing the oversmoothing issue, enabling VIT models to achieve greater depth and better robustness.

**Strengths:**

1. The authors undertake a comprehensive investigation, encompassing both theoretical and empirical analyses, to gain insights into the oversmoothing phenomenon within Transformers. They give clear explanations regarding the underlying reasons for oversmoothing.

2.  A novel reparameterization technique is introduced as a solution to mitigate the oversmoothing problem.

3. Empirical evaluations validate the efficacy of the proposed approach in addressing oversmoothing, showcasing its ability to deepen Transformers and enhance robustness in Transformer models.

**Weaknesses:**

The empirical evaluations in this study are exclusively conducted on computer vision tasks. However, there is an expectation for a broader and more diverse range of tasks, including but not limited to natural language processing (NLP) and multimodal tasks, to provide a more comprehensive evaluation of the proposed approach.

**Questions:**

Can the proposed approach effectively mitigate oversmoothing even when employing an exceptionally high number of layers?
Additionally, what is the performance impact of this approach on NLP tasks?

---

> ### Author Response · Authors · 2023-11-18
> **Author Response to Reviewer m7iA**
>
> Dear Reviewer m7iA,
>
> Thank you so much for your time and feedback. We respond to each question below.
>
> > [..Weaknesses:..]
>
> > [..an expectation for a broader and more diverse range of tasks...what is the performance impact of this approach on NLP tasks?..]
>
> We definitely aim to investigate this in future work. In large part, most research we are aware of on Transformer oversmoothing divides into either vision-focused or NLP-focused work, both of which are usually entirely empirical (with the rare exceptions of Dong et al., 2021 and Wang et al., 2022). Because our theory extends the theoretical work of Dong et al., 2021 and Wang et al., 2022 who restrict their focus to vision Transformers, we originally opted to focus solely on vision as well.
>
> This said, we agree that it would strengthen the paper to extend the evaluation to another domain. We are currently running an NLP experiment that should be ready on Monday. We will get back to you with these results as soon as possible!
>
> > [..Questions..]
>
> > [..Can the proposed approach effectively mitigate oversmoothing even when employing an exceptionally high number of layers?..]
>
> This is a good question! To answer this we have added an experiment where we increase the depth of the ViT-Ti models on CIFAR100. We have also added a parameterization variant where the first L/2 layers sharpen (as in ViT-Ti$^-$) and the remaining $L/2$ layers are left normal (as in ViT-Ti), which we call ViT-Ti*. Finally we have added a comparison FeatScale by Wang et al., 2022. All results are averaged over 3 trials with standard deviations (we bold the best result in each column and any method whose mean plus standard deviation is greater than or equal to mean of the best result):
>
> [new] Increased Depth (Table 3)
>
> | Depth       	| 12       	| 24       	|
> |----------------|--------------|--------------|
> | ViT-Ti     	| 66.78 ± 0.1  | 67.22 ± 0.1  |
> | ViT-Ti + FeatScale | 66.57 ± 0.5  | 67.71 ± 0.5  |
> | ViT-Ti$^+$ 	| 64.34 ± 0.3  | 65.65 ± 0.3 |
> | ViT-Ti$^-$ 	| 66.62 ± 0.3  | 66.76 ± 0.3  |
> | ViT-Ti$^*$ 	| **67.84 ± 0.6**  | **69.05 ± 0.1**  |
>
> Here ViT-Ti$^+$ and ViT-Ti$^-$ both underperform ViT-Ti, and we suspect this is because ViT-Ti$^-$ sharpens too much, amplifying noise in the data (whereas ViT-Ti$^+$ oversmooths the data). On the other hand ViT-Ti* significantly outperforms all other models not just for larger depths in Table 3, but also in corruption robustness (Table 4), and data efficiency (Table 2). This suggests that both oversmoothing and oversharpening are detrimental to performance, and that a careful balance is key to an accurate model. We have updated the paper to describe these results and look forward to investigating this further in future work.
>
> Thank you again for your time. Let us know if you have any other questions and we will respond as soon as possible.

---

> ### Author Response · Authors · 2023-11-21
>
> > [..an expectation for a broader and more diverse range of tasks...what is the performance impact of this approach on NLP tasks?..]
>
> We are still running NLP experiments and hope to have it by tomorrow

---

> ### Author Response · Authors · 2023-11-22
> **Author Comment to Reviewer m7iA**
>
> Dear Reviewer m7iA,
>
> Thank you again for your time on the paper. Do you have any extra questions about the paper or our response?
>
> Let us know and we will try to response as soon as possible before the discussion period ends on November 22.
>
> Thank you,
>
> The authors

---

### Official Review · Reviewer_QJ8g · 2023-11-01

**Soundness:** 4 excellent
**Presentation:** 2 fair
**Contribution:** 3 good
**Rating:** 5
**Confidence:** 4

**Summary:**

This paper investigates how Transformers oversmooth through the lens of low/high-pass filters and introduces a reparameterization of the product of the output and value projection matrices to avoid oversmoothing. The paper presents conditions in which Transformers are not low-pass filters using analysis of the domination of the eigenvalues of the attention matrix and how at the infinite-layer limit, the feature vectors converge to different solutions based on the domination of these eigenvalues.

**Strengths:**

1. The paper provides a good analysis of when and how to avoid over-smoothing in Transformers is an important problem since Transformers in practice have many layers.

2. The paper is well-motivated.

**Weaknesses:**

1. The proposed method for avoiding oversmoothing is based on the claim that H becomes more symmetric as the training progresses. However, it is not confirmed that H eventually becomes a symmetric matrix. Therefore, parameterizing H as a symmetric matrix might restrict the expressive power of the Transformer model as a whole.

2. There is no definite answer to why the ViT-Ti^- version improves robustness and data efficiency. The intuition provided by the authors is not enough to answer this because robustness and data efficiency are more nuanced than simply choosing higher-frequency features.

3. The writing of the paper is not polished, which creates confusion in key details of the paper. For example, given the redundancy in the cases in Theorem 1, maybe Theorem 1 could be rewritten to make the main result easier to read.

4. In Table 1, the authors show the distribution of the dominating eigenvalues but do not show whether the features oversmooth when \lambda_1^A dominates. This must be verified because the Theorems presented only work when the attention matrix A is fixed across layers, which does not hold for all practical settings. Therefore, showing the distribution of the dominating eigenvalues without showing how they affect the final features is vacuous.

**Questions:**

1. The precise definition of dominating eigenvalues should be given to make Theorem 1 easier to read.

2. “To show a case where over-smoothing is guaranteed, we also define a model diag(ΛH) := −(ψ^2), which we refer to using the superscript +.” I think the authors mean diag(ΛH) := +(ψ^2).

Minor Comments that did not affect the score:

1. It would be good if the authors could compare the runtime of their ViT-Ti^+ with other methods.

2. The authors showed the distribution of dominating eigenvalues of their - version on CIFAR10, but not on ImageNet. It would be interesting to see if the distribution of the dominating eigenvalues is still the same on ImageNet.

---

> ### Author Response · Authors · 2023-11-18
> **Author Response to Reviewer QJ8g (Part 1)**
>
> Dear Reviewer QJ8g,
>
> Thank you so much for your time and questions. We respond to each of them below.
>
> > [..Weaknesses:..]
>
> > [..1. ...parameterizing H as a symmetric matrix might restrict the expressive power..]
>
> Thanks for this comment. Our original motivation for using a symmetric parameterization was inspired by [Hu et al., 2019. Exploring weight symmetry in deep neural networks.] who show that weight symmetry in NNs does not affect their universal approximation capabilities. However, we agree that it would be interesting to test a non-symmetric parameterization. Based on your suggestion we have implemented a new parameterization for $\mathbf{H}$ as $\mathbf{H} = \mathbf{V}_H \Lambda \mathbf{V}_H^{-1}$. Specifically, we take gradients to $\mathbf{V}_H$ in the backwards pass, and compute $\mathbf{H}$ using the above equation in the forwards pass. For both the symmetric and non-symmetric versions we test out 2 variants: 1. (-) i.e., the sharpening model: $\mbox{diag}(\Lambda_H ) := −(\psi^2)$ and a new variant 2. (*) where the first L/2 layers are parameterized as the sharpening model above, and the remaining L/2 layers are left normal (as in ViT-Ti). We use the suffix “-NS” for the non-symmetric parameterizations. We have also included a new baseline, FeatScale [Wang et al., 2022], as well as a new experiment where we double the depth of the network. Here are the new accuracy results, averaged over 3 trials with standard deviations (we bold the best result in each column and any method whose mean plus standard deviation is greater than or equal to mean of the best result):
>
> Data Efficiency (Table 2)
>
> | Data Kept          	| 100%     	| 50%      	| 10%      	|
> |------------------|--------------|--------------|--------------|
> | ViT-Ti       	| 66.78 ± 0.1  | 56.38 ± 0.5  | 32.47 ± 0.9  |
> | ViT-Ti + FeatScale | 66.57 ± 0.5  | 57.08 ± 0.3  | 32.50 ± 0.4  |
> | ViT-Ti$^+$   	| 64.34 ± 0.3  | 55.21 ± 0.5  | 28.67 ± 0.5  |
> | ViT-Ti$^-$   	| 66.62 ± 0.3  | 56.58 ± 0.3  | **33.99 ± 0.4**  |
> | ViT-Ti$^*$   	| 67.84 ± 0.6  | **57.78 ± 0.4**  | **33.64 ± 0.5**  |
> | ViT-Ti-NS$^-$	| 67.80 ± 0.2  | 57.54 ± 0.1  | **33.78 ± 0.3**  |
> | ViT-Ti-NS$^*$	| **68.53 ± 0.6**  | **57.67 ± 0.3**  | 33.75 ± 0.2  |
>
>
> [new] Increased Depth (Table 3)
>
> | Depth       	| 12       	| 24       	|
> |----------------|--------------|--------------|
> | ViT-Ti     	| 66.78 ± 0.1  | 67.22 ± 0.1  |
> | ViT-Ti + FeatScale | 66.57 ± 0.5  | 67.71 ± 0.5  |
> | ViT-Ti$^+$ 	| 64.34 ± 0.3  | 65.65 ± 0.3 |
> | ViT-Ti$^-$ 	| 66.62 ± 0.3  | 66.76 ± 0.3  |
> | ViT-Ti$^*$ 	| 67.84 ± 0.6  | 69.05 ± 0.1  |
> | ViT-Ti-NS$^-$  | 67.80 ± 0.2  | 67.09 ± 0.1  |
> | ViT-Ti-NS$^*$  | **68.53 ± 0.6**  | **69.64 ± 0.3**  |
>
>
> Corruption Robustness (Table 4)
>
> | Corruption Intensity | 0          	| 1          	| 2          	| 3          	| 4          	| 5          	|
> |----------------------|----------------|----------------|----------------|----------------|----------------|----------------|
> | ViT-Ti           	| 66.78 ± 0.1	| 61.59 ± 0.2	| 57.75 ± 0.2	| 53.12 ± 0.1	| 49.32 ± 0.1	| 39.17 ± 0.2	|
> | ViT-Ti + Featscale   | 66.57 ± 0.5	| 61.25 ± 0.5	| 57.36 ± 0.6	| 52.64 ± 0.6	| 48.87 ± 0.5	| 38.77 ± 0.4	|
> | ViT-Ti$^+$       	| 64.34 ± 0.3	| 58.36 ± 0.4	| 53.83 ± 0.5	| 49.03 ± 0.6	| 45.15 ± 0.6	| 35.86 ± 0.6	|
> | ViT-Ti$^-$       	| 66.62 ± 0.3	| 61.62 ± 0.2	| 58.00 ± 0.3	| 53.56 ± 0.2	| 49.70 ± 0.3	| 40.07 ± 0.2	|
> | ViT-Ti$^*$       	| 67.84 ± 0.6	| 62.49 ± 0.5	| 58.82 ± 0.7	| 54.32 ± 0.7	| 50.40 ± 0.6	| 40.38 ± 0.7	|
> | ViT-Ti-NS$^-$    	| 67.80 ± 0.2	| **63.03 ± 0.2**	| **59.65 ± 0.1**	| **55.05 ± 0.3**	| **51.30 ± 0.1**	| **41.33 ± 0.1**	|
> | ViT-Ti-NS$^*$    	| **68.53 ± 0.6**	| 63.01 ± 0.1	| **59.44 ± 0.3**	| **54.90 ± 0.4**	| **51.12 ± 0.4**	| 40.90 ± 0.4	|
>
> We have added these results to the paper. The non-symmetric parameterization does improve upon the symmetric parameterization in some cases (particularly for increased depth and corrupted data). We appreciate your suggestion to investigate this.
>
>
> > [..2. ...robustness and data efficiency...]
>
> Thanks. This seems to be a very deep question. We originally suspected a link between sharpening and data efficiency because of the eigenvalue distribution of the Data Efficient Image Transformer (DeiT-B/16) in Table 1: compared to all other baseline models, it has the highest distribution of sharpening eigenvalues. This is what led us to test data efficiency. We also hypothesized that it may help with input corruption robustness as many of the corruption techniques in Hendrycks & Dietterich, 2019 blur or reduce the resolution of the image: any further smoothing would likely make classification even harder. As far as we are aware we are the first to investigate the relationship between sharpening and data efficiency and corruption robustness. We agree this is interesting to further investigate. If you have any suggestions for additional experiments we will try to implement them as quickly as possible.
>
> (response continues below)

---

> ### Author Response · Authors · 2023-11-18
> **Author Response to Reviewer QJ8g (Part 2)**
>
> > [..3. ...redundancy in the cases in Theorem 1..]
>
> Thank you for pointing out this redundancy! We have fixed this.
>
> > [..4. ...the authors show the distribution of the dominating eigenvalues but do not show whether the features oversmooth when $\lambda_1^A$ dominates...]
>
> We believe there is a small confusion here. For ViT-Ti$^+$ we have that (a) Figure 3 shows that the features do oversmooth more and more, and (b) Table 1 shows that $\lambda_n^A$ dominates. On the other hand, for ViT-Ti$^-$ we have that (a) Figure 3 indicates that the features do not oversmooth, and (b) Table 1 shows that $\lambda_1^A$ dominates. This shows that even when the attention matrix $\mathbf{A}$ is not fixed across layers the results agree with the Theorem statements. Thank you for letting us clarify this, we will add these details to the text to make this clearer.
>
> > [..Questions..]
>
> > [..1. The precise definition of dominating eigenvalues should be given...]
>
> We agree. We have pulled this definition out of the proof and placed it into the main paper.
>
> > [..2.  I think the authors mean $\mbox{diag}(\Lambda_H ) := +(\psi^2)$..]
>
> Thank you for catching this! We have fixed this.
>
> > [..Minor Comments..]
>
> > [..1. compare the runtime of their ViT-Ti$^+$ with other methods...]
>
> Thanks. The training time of ViT-Ti versus our original proposed methods is as follows (averaged over 3 runs):
> |     |  Training time |
> |----------|---------------|
> | ViT-Ti  |   5h 34min |
> | ViT-Ti$^+$|  6h 03min |
> | ViT-Ti$^-$ |  6h 03min |
>
> The throughput of each method (in images/second, averaged over 10 runs) is:
>
> |     |  Throughput |
> |----------|---------------|
> | ViT-Ti  |   1683.0 |
> | ViT-Ti$^+$|  1642.6 |
> | ViT-Ti$^-$ |  1639.0 |
>
> We will add these timings to the text.
>
> > [..2. The authors showed the distribution of dominating eigenvalues of their - version on CIFAR10, but not on ImageNet...]
>
> We think there is a small confusion. The dominating eigenvalue distribution of our $\mbox{}^-$ version is always 0%: $(1 + \lambda^H_j \lambda^A_n)$ and 100%: $(1 +  \lambda^H_j \lambda^A_1)$, regardless of the data, as we control the eigenvalues via bounding $\Lambda^H$ in our parameterization. We will clarify this in the paper.
>
> Thank you again for your time. Let us know if you have any other questions and we will respond as soon as possible.

---

> ### Author Response · Authors · 2023-11-22
> **Author Comment to Reviewer QJ8g**
>
> Dear Reviewer QJ8g,
>
> Thank you again for your time on the paper. Do you have any extra questions about the paper or our response?
>
> Let us know and we will try to response as soon as possible before the discussion period ends on November 22.
>
> Thank you,
>
> The authors

---

### Meta-Review · Area_Chair_SLNX · 2024-01-02

**Metareview:**

The paper attempts to improve our understanding of working of transformers. In particular, the authors challenge a notion that transformers inherently act as low-pass filters introduced by a few prior works e.g. Wang et al. (2022). The paper shows transformers can over-smooth or not depending on eigenvalues of attention and weights under some assumptions. Furthermore, authors leverage this observation to reparametrize transformer that avoids over-smoothing. Some empirical studies were carried out on vision tasks in the submission using complete transformer architecture including the layer-norms. Unfortunately the submission received a lukewarm response from the reviewers raising many potential concerns from assumptions being realistic to limited experimental evaluation and comparisons to writing quality. While the contribution of the paper are clear, the presentation and wiriting can be further polished to make the flow and logical reasoning better, e.g. while the applications of low-data and corruption interesting, their motivation for this study is less clear. Other open questions: as results are for limiting case, it might not imply much for what will happen for a fixed small L. In figure 3 all have decreasing trend with L (except for the forced ones). Would it be more helpful to include experiments with varying L? Moreover, theoretical exposition in the paper is carried out assumping shared weight across layers, so maybe conducting experiments with shared weight transformer models like ALBERT to align better with theory? We thank the authors for actively engaging during the discussion phase to improve the paper by providing new experimental results on 24 layer and NLP task of next word prediction, but on non-standard configuration which makes it hard to compare and understand contribution of the paper. We are empathetic with the authors as the paper seems timely, but none of the reviewers were willing to champion for the paper given the shortcomings. We highly encourage authors to improve the work based on reviewer suggestion and submit to next venue.


Minor:
Typo: in theorem 3, the equation number of case 3, should be eq 7?

**Justification For Why Not Higher Score:**

Limited Scope of Evaluation: The empirical analysis is primarily on vision tasks, although one non-standard NLP task was added during discussion phase. Also experiments more aligned with theory can be useful, e.g. in theory weights are shared across layers, so experiments surrounding it can be interesting.
Comparative Analysis: The paper could benefit from more comprehensive comparisons with other techniques addressing oversmoothing like layer-norm.
Better presentation: Some parts of the theoretical analysis remain unecessarily complex and could be made more accessible.

**Justification For Why Not Lower Score:**

N/A

---

### Decision · Program_Chairs · 2024-01-16

Reject